# CD8+ T cell landscape in Indigenous and non-Indigenous people restricted by influenza mortality-associated HLA-A*24:02 allomorph

Luca Hensen [1,26], Patricia T. Illing [2,26], E. Bridie Clemens[1,26], Thi H. O. Nguyen [1], Marios Koutsakos[1], Carolien E. van de Sandt [1,3], Nicole A. Mifsud [2], Andrea T. Nguyen[2], Christopher Szeto [2], Brendon Y. Chua[1], Hanim Halim[2], Simone Rizzetto [4], Fabio Luciani[4], Liyen Loh [1], Emma J. Grant[1,2], Phillipa M. Saunders [1], Andrew G. Brooks [1], Steve Rockman[1,5], Tom C. Kotsimbos[6,7], Allen C. Cheng[8,9], Michael Richards[10], Glen P. Westall[11], Linda M. Wakim[1], Thomas Loudovaris[12], Stuart I. Mannering [12], Michael Elliott[13,14], Stuart G. Tangye [15,16], David C. Jackson [1], Katie L. Flanagan [17,18,19,20], Jamie Rossjohn [2,21,22], Stephanie Gras[2,22,23], Jane Davies[24], Adrian Miller[25], Steven Y. C. Tong [10,24,27], Anthony W. Purcell [2,27] & Katherine Kedzierska [1,27✉]

Indigenous people worldwide are at high risk of developing severe influenza disease. HLA-A*24:02 allele, highly prevalent in Indigenous populations, is associated with influenza-induced mortality, although the basis for this association is unclear. Here, we define CD8+ T-cell immune landscapes against influenza A (IAV) and B (IBV) viruses in HLA-A*24:02-expressing Indigenous and non-Indigenous individuals, human tissues, influenza-infected patients and HLA-A*24:02-transgenic mice. We identify immunodominant protective CD8+ T-cell epitopes, one towards IAV and six towards IBV, with A24/PB2$_{550-558}$-specific CD8+ T cells being cross-reactive between IAV and IBV. Memory CD8+ T cells towards these specificities are present in blood (CD27+CD45RA− phenotype) and tissues (CD103+CD69+ phenotype) of healthy individuals, and effector CD27−CD45RA−PD-1+CD38+CD8+ T cells in IAV/IBV patients. Our data show influenza-specific CD8+ T-cell responses in Indigenous Australians, and advocate for T-cell-mediated vaccines that target and boost the breadth of IAV/IBV-specific CD8+ T cells to protect high-risk HLA-A*24:02-expressing Indigenous and non-Indigenous populations from severe influenza disease.

A full list of author affiliations appears at the end of the paper.

Newly emerging respiratory viruses pose a major global pandemic threat, leading to significant morbidity and mortality, as exemplified by 2019 SARS-CoV2, avian influenza H5N1 and H7N9 viruses, and the 1918–1919 H1N1 pandemic catastrophe. Influenza A viruses (IAV) can cause sporadic pandemics when a virus reassorts and rapidly spreads across continents, causing millions of infections and deaths[1]. Additionally, seasonal epidemics caused by co-circulating IAV and influenza B viruses (IBV) result in 3–5 million cases of severe disease and 290,000–650,000 deaths annually[2,3]. Severe illness and death from seasonal and pandemic influenza occur disproportionately in high-risk individuals, including Indigenous populations. This is most evident when unpredicted seasonal or pandemic viruses emerge in human circulation. During the 1918–1919 influenza pandemic, 100% of Alaskan adults died in some isolated villages, while only school-aged children survived[4]. Western Samoa was the hardest hit with a total population loss of 19–22%[5]. As many as 10–20% of Indigenous Australians died from pandemic influenza in 1919[6] in comparison to <1% of other Australians, with some reports showing up to 50% mortality in Indigenous Australian communities[7].

During the 2009 A/H1N1 influenza pandemic, Indigenous populations worldwide were more susceptible to influenza-related morbidity and mortality. Hospitalization and morbidity rates were markedly increased in Indigenous Australians[8,9], with 16% of hospitalized pandemic H1N1 (pH1N1) patients in Australia being Indigenous. The relative risk for Indigenous Australians compared to non-Indigenous Australians for hospitalization, ICU admission, or death was 6.6, 6.2, or 5.2 times higher, respectively[10]. This was mirrored in Indigenous populations globally, including American Indians and Alaskan Native people (4-fold higher mortality rate compared to non-Indigenous Americans)[11], native Brazilians (2-fold higher hospitalization rate)[12], New Zealand Maori (5-fold higher hospitalization rate), and Pacific Islanders (7-fold higher hospitalization rate)[13,14]. Although the impact of influenza pandemics is more pronounced in Indigenous populations globally, these disproportionate hospitalization rates also occur during seasonal infections. During 2010–2013, Indigenous Australians had increased influenza-related hospitalizations across all age groups (1.2–4.3-fold higher compared to non-Indigenous)[15]. Indigenous populations, especially Australians and Alaskans, are also predicted to be at greater risk from severe disease caused by the avian-derived H7N9 influenza virus, with mortality rates being >30% and hospitalization >99% in China[16]. While higher influenza infection rates could relate to overcrowded living conditions, increased severity and prolonged hospitalization most likely reflect differences in pre-existing immunity that facilitates recovery. However, the underlying immunological and host factors that account for severe influenza disease in Indigenous individuals are far from clear.

Antibody-based vaccines towards variable surface glycoproteins, hemagglutinin (HA), and neuraminidase (NA), are an effective way to combat seasonal infections, yet they fail to provide effective protection when a new, antigenically different IAV emerges[17]. In the absence of antibodies, recall of pre-existing cross-protective memory CD8[+] T-cells minimizes the effects of a novel IAV, leading to a milder disease after infection with distinct strains[18–23]. Such pre-existing memory CD8[+] T-cells provide broadly heterotypic or cross-reactive protection and can recognize numerous IAV, IBV, and influenza C viruses capable of infecting humans[24], promoting rapid host recovery. During the 2013 H7N9 IAV outbreak in China, recovery from severe H7N9 disease was associated with early CD8[+] T-cell responses[21,25]. Patients discharged early after hospitalization had early (day 10) robust H7N9-specific CD8[+] T-cells responses, while those with prolonged hospital stays showed late (day 19) recruitment of CD8[+] and CD4[+] T-cells. Thus, with the continuing threat of unpredicted influenza strains, there is a need for targeting cellular immunity that provides effective, long-lasting, and cross-strain protective immunity, especially for high-risk groups such as Indigenous populations. However, despite the heavy burden of disease in Indigenous communities, there is scant data on immunity to influenza viruses in Indigenous populations from around the world.

As CD8[+] T-cell recognition is determined by the spectrum of human leukocyte antigens (HLAs) expressed in any individual, and HLA profiles differ across ethnic groups, defining T-cell epitopes restricted by HLAs predominant in some Indigenous populations is necessary to understand pre-existing CD8[+] T-cell immunity to influenza. We previously analyzed the HLA allele repertoire in Indigenous Australians[26] and found that HLA-A*24:02 (referred to as HLA-A24 hereafter), an HLA associated with influenza-induced mortality during the 2009-pH1N1 outbreak[27], is the second most prominent HLA in Indigenous Australians[26,28]. HLA-A24 is also common to other Indigenous populations highly affected by influenza[16]. Thus, analysis of prominent influenza-specific CD8[+] T-cell responses restricted by HLA-A24 is needed to understand the relationship between this allele and disease susceptibility. These specificities will also inform strategies to prime effective T-cell immunity in vulnerable communities.

Here, we define CD8[+] T-cell immune landscapes against IAV and IBV, restricted by the mortality-associated HLA-A24 allomorph. We identify IAV- and IBV-specific HLA-A24 immunopeptidomes and screen immunogenicity of novel peptides in HLA-A24-expressing mice, peripheral blood of Indigenous and non-Indigenous HLA-A24[+] healthy and influenza-infected individuals, and human tissues. Our studies provide evidence of the breadth of influenza-specific CD8[+] T-cell specificities restricted by a mortality-associated risk allomorph HLA-A24. These findings have implications for the incorporation of key CD8[+] T-cell targets in a T-cell-mediated vaccine to protect Indigenous people globally from unpredicted influenza viruses.

## Results

**High prevalence of HLA-A24 expression in Indigenous populations.** As HLA-A24 has been linked to pH1N1-related mortality[27], we first determined HLA-A24 distribution in Indigenous and non-Indigenous populations worldwide using the published allele frequency database. Compared to the 10% global distribution of HLA-A24, the detected frequencies of HLA-A24 were the highest in Oceania (37%), North-East Asia (32.9%), Australia (21.4%), and Central and South America (20.6%) (Fig. 1a). This was mainly due to a particularly high HLA-A24 prevalence in Indigenous populations in those regions, especially in the Pacific. HLA-A24 was highly prevalent in Indigenous Taiwan Paiwan (96.1%), Papa New Guinea Karimui Plateau Pawaia (74.4%), New Caledonia (60.7%), Alaskan Yupik (58.1%), New Zealand Maori (38%), American Samoans (33%), Chile Easter Island (35.8%), and some Australian Indigenous people (24%) (Fig. 1a), which highlights its key importance in shaping CD8[+] T-cell immunity in Indigenous populations.

To define influenza-specific CD8[+] T-cell responses in Indigenous Australians, we recruited 127 participants from the Northern Territory, Australia into the LIFT (Looking Into influenza T-cell immunity) cohort[26]. The mean age of the participants was 39 years, with a standard deviation of 14 years and 58% of female participants. 36% of the LIFT donors expressed at least one HLA-A24 allele, with 33% of those being HLA-A24 homozygous (Table 1). Notably, HLA-A24 was most commonly expressed with HLA-A*11:01, -A*34:01, -B*13:01,

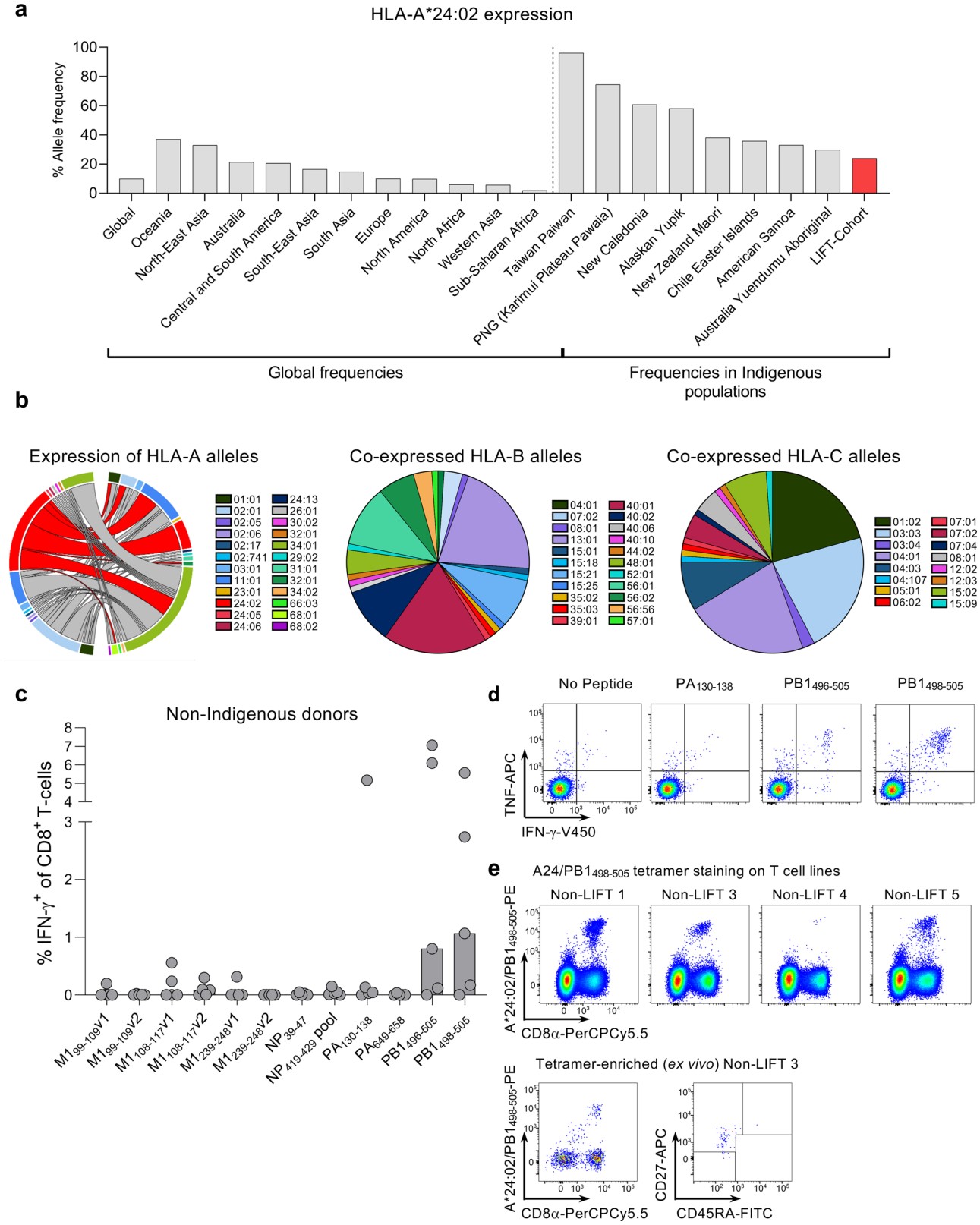

**Fig. 1 HLA-A24 expression and characterization of published IAV epitopes. a** Allele frequency of HLA-A*24:02 according to geographic region (left panel) and Indigenous population in the pacific region (right panel) (source www.allelefrequencies.net accessed 20/01/2020). **b** Co-expression of HLA-A, B and C alleles with the HLA-A*24:02 allele in the LIFT cohort, where Circos plot is shown for HLA-A and pie charts are shown for HLA-C and B alleles that are co-expressed with the HLA-A*24:02 allele. **c** Screening of previously identified IAV HLA-A*24:02 peptides via IFN-γ-ICS assay following peptide expansion of PBMCs from 5 non-Indigenous individuals (median bar graphs are shown). **d** Representative FACS plots of cytokine expression for immunogenic peptides. **e** A24/PB1$_{498-505}$-tetramer staining of peptide-expanded T-cell lines (top panels) and after ex vivo TAME from non-LIFT 3 donor (bottom panels) showing the TAME-enriched population and phenotype.

**Table 1 Summary LIFT cohort.**

| Looking into influenza T cell immunity (LIFT) cohort | |
|---|---|
| Individuals (n) | 127 |
| Age (mean ± SD) | 39 ± 14 |
| Females (%) | 58 |
| HLA-A24⁺ individuals (%) | 36 |

**Table 2 Published immunogenic A*24:02-restricted IAV peptides.**

| Peptide | Sequence | References |
|---|---|---|
| M1₉₉₋₁₀₉ | LYRKLKREITF | Assarsson et al.[29] |
| | LYKKLKREITF[a] | Zhao et al.[30] |
| | LYKKLKREMTF[b] | Zhao et al.[30] |
| M1₉₉₋₁₀₇ | LYKKLKREI | Liu et al.[31] |
| M1₁₀₈₋₁₁₇ | TFHGAKEVSL | Liu et al.[31] |
| M1₂₃₉₋₂₄₈ | AYQKRMGVQM | Liu et al.[31] |
| NP₃₉₋₄₇ | FYIQMCTEL | Alexander et al.[32] |
| NP₄₁₉₋₄₂₉ | PFERATVMAAF | Liu et al.[31] |
| PA₁₃₀₋₁₃₈ | YYLEKANKI | Alexander et al.[32] |
| PA₆₄₉₋₆₅₈ | LYASPQLEGF | Liu et al.[31] |
| PB1₄₉₆₋₅₀₅ | FYRYGFVANF | Alexander et al.[32] |
| PB1₄₉₈₋₅₀₅ | RYGFVANF | Liu et al.[31] |

[a]Sequence found in pH1N1 and H5N1.
[b]Sequence found in H7N9 and H9N2. Not tested in this study.
Underline indicates position of amino acid substitutions compared to the first mentioned sequence.

-B*15:21, -B*40:01, -B*40:02, -B*56:01, -C*04:03, -C*03:03, -C*04:01, and -C*04:03 in Indigenous Australians and were less expressed with alleles common in Caucasian populations, such as HLA-A*01:01, -A*02:01, -B*07:02, and -B*08:01 (Fig. 1b).

**CD8⁺ T-cell responses towards published IAV-specific HLA-A24-epitopes.** A handful of IAV-specific HLA-A24-restricted epitopes have been described[29–32] (Table 2). We aimed to validate the immunogenicity of these epitopes by probing memory CD8⁺ T-cells within peripheral blood mononuclear cells (PBMCs) of healthy non-Indigenous HLA-A24-expressing donors (Supplementary Table 1) using an in vitro peptide stimulation assay (Fig. 1c–e). Only 3 out of 12 peptides (PB1₄₉₈₋₅₀₅, PB1₄₉₆₋₅₀₅, PA₁₃₀₋₁₃₈) induced CD8⁺ T-cell proliferation and IFN-γ/TNF production in a limited number of donors (3/5, 3/5, and 1/5, respectively) (Fig. 1c, d). We deduced that the minimal epitope for PB1₄₉₆₋₅₀₅ responses came from the PB1₄₉₈₋₅₀₅ epitope. The specificity of PB1₄₉₈₋₅₀₅-specific CD8⁺ T-cell responses, observed in multiple donors, was further verified by A24/PB1₄₉₈₋₅₀₅ tetramer staining on both in vitro-cultured A24/PB1₄₉₈₋₅₀₅ CD8⁺ T-cell lines and A24/PB1₄₉₈₋₅₀₅⁺CD8⁺ T-cells detected directly ex vivo by tetramer enrichment (Fig. 1e). Thus, while 3 of the previously published peptides elicited IFN-γ responses in a selected number of HLA-A24-expressing individuals, we sought to determine whether as yet unidentified epitopes might provide more robust IAV-specific CD8⁺ T-cell responses in HLA-A24-expressing individuals.

**Identification of novel HLA-A24-restricted IAV and IBV epitopes.** To identify new A24/influenza-derived epitopes, we utilized an immunopeptidomics approach to sequence HLA-bound peptides on influenza-infected cells by liquid chromatography-tandem mass spectrometry (LC–MS/MS)[24]. Experiments were performed with the class I-reduced C1R B-lymphoblastoid cell line (minimal HLA-B*35:03 surface expression but normal levels of HLA-C*04:01

expression[33]) and an HLA-A*24:02-transfected C1R cell line (C1R.A24)[34]. Initial experiments were performed in both cell lines at 16 h post-infection with and without the HKx31 IAV virus, isolating peptide-bound HLA-I molecules post-lysis utilizing the pan HLA-I antibody W6/32 (Supplementary Data 1). In subsequent experiments, to improve the confidence of assignment of binding to HLA-A24, HLA-C*04:01 depletion of the lysates was performed with the HLA-C-specific antibody DT9 prior to isolation of the remaining HLA-I (A*24:02»B*35:03) with W6/32 antibody. Utilizing this workflow[35], we assessed peptide presentation in C1R.A24 cells at 2, 4, 8, and 12 h post-infection with either A/HKx31 or B/Malaysia/2506/2004 viruses. Analyses were restricted to up to 12 h post-infection due to observations of marked HLA-I downregulation at 16 h post-infection[36].

In total, 12 immunopeptidome data sets containing HLA-A24-restricted peptides were generated including 3 from uninfected C1R.A24 cells, 5 from HKx31-infected, and 4 from B/Malaysia-infected C1R.A24 cells (Supplementary Fig. 1a and Supplementary Data 1). An additional 3 data sets for endogenous HLA-I of C1R cells (C1R W6/32 isolation of HLA-B*35:03 and HLA-C*04:01 after 16 h HKx31 infection; C1R.A24 - DT9 isolation of HLA-C*04:01 from uninfected cells and after 12 h HKx31 infection) and 2 data sets for endogenous HLA-II (C1R.A24: HLA-DR12, -DPB1*04:01,04:02 and -DQ7 from uninfected and 12 h HKx31 infection) were also generated as comparators to help establish true HLA-A24 binders (Supplementary Fig. 1b, c). Comparisons to previously identified B/Malaysia-derived HLA ligands for C1R were also used[24] to distinguish HLA-B*35:03 and HLA-C*04:01 binding peptides from those binding to HLA-A*24:02. Across the 12 HLA-A24 data sets, a total of 9051 non-redundant peptide sequences were assigned to the human proteome at a 5% false discovery rate (FDR). As expected for HLA-I ligands, the majority of peptides were 9–11 amino acids in length but dominated by 9mers (Fig. 2a). Consistent with the HLA-A24 peptide-binding motif generated by NetMHC4.0[37,38] motif viewer, enrichment of Tyr/Phe at P2 and Phe/Leu/Ile at P9 were observed (Fig. 2b). Peptides binding the endogenous HLA-I of C1R were not removed in this analysis due to the similar preference of HLA-C*04:01 for 9mer peptides possessing Phe/Tyr at P2 and Phe/Leu at P9, which may result in shared ligands (Supplementary Fig. 1d). To maximize the identification of potential virus-derived peptides, assignments to the viral proteome or 6-frame translation of the viral genome were considered without an FDR cut-off. Instead, lack of appearance in uninfected data sets and predicted binding affinity (NetMHCpan4.0[39–41]) for HLA-A24 were used to determine likely candidate epitopes. Thus, 52 HKx31-derived and 48 B/Malaysia-derived peptides were identified as potential HLA-A24-restricted epitopes (Fig. 2c), of which 26 IAV-derived and 29 IBV-derived peptides were identified at a 5% FDR (Supplementary Data 1). The identified peptides spanned the viral proteomes including frame-shift proteins, representing 6 IAV proteins and 9 IBV proteins (Fig. 2d, e). Interestingly, most HKx31-derived ligands mapped to PB2 > PB1 > HA viral proteins and none were observed from NA or M1, while B/Malaysia-derived ligands predominantly mapped to NP/HA > NA. During the time course analyses, broadest peptide identification was achieved for both viruses between 8 and 12 h post-infection, while no influenza-derived peptides were identified at 2 h post-infection, and those identified at 4 h were of lower confidence (Fig. 2f, g and Supplementary Data 1). 10 potential HKx31-derived ligands were also identified for each of HLA-B*35:03 and HLA-C*04:01 based on predicted binding and/or appearance in control data sets (Supplementary Fig. 1e and Supplementary Data 1). Furthermore, analysis of the peptides presented by HLA-II molecules showed domination of the virus-derived immunopeptidome by HA (Supplementary Fig. 1f and Supplementary Data 1), as previously observed for IBV[24]. A

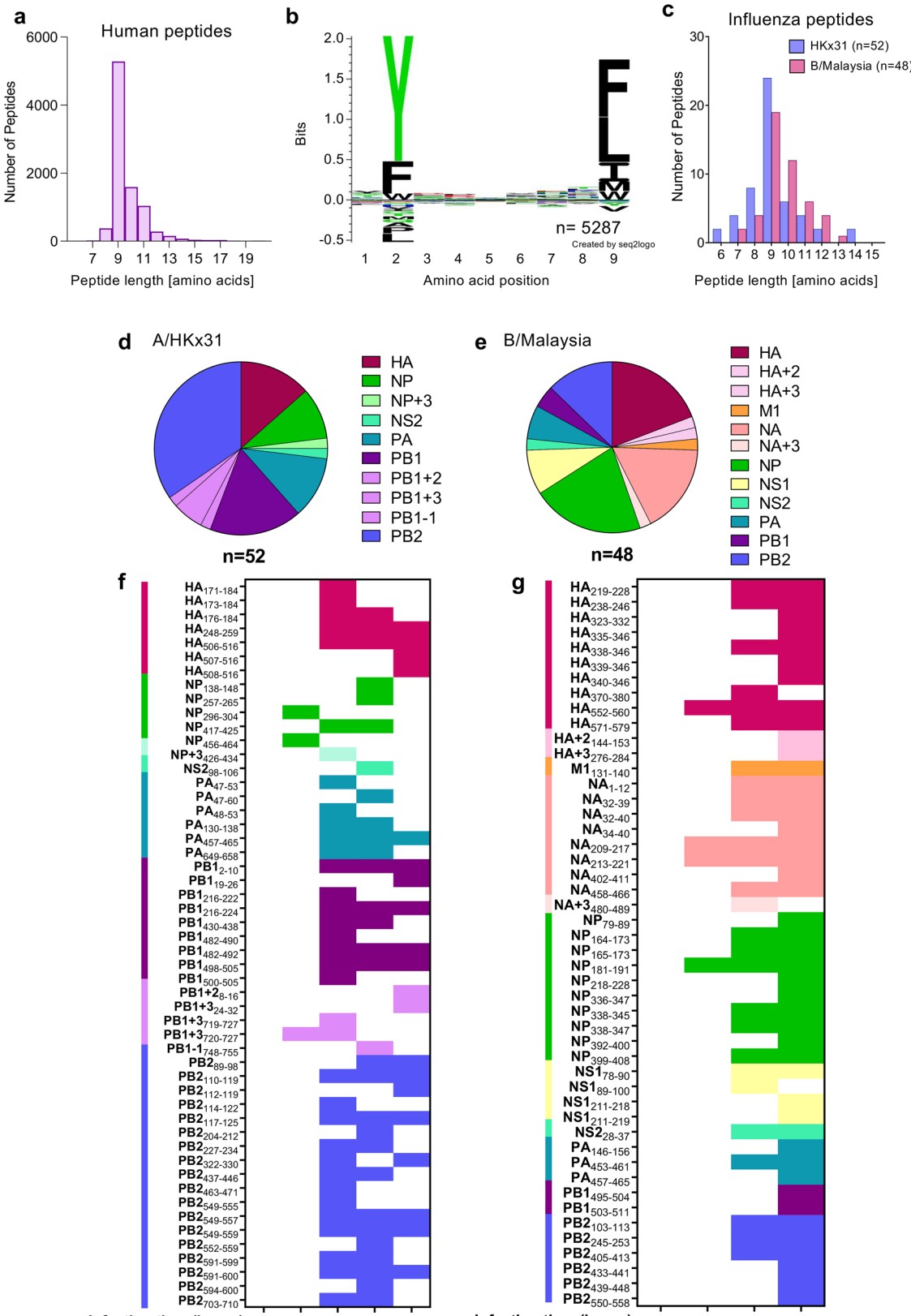

selection of 48 IAV and 41 IBV peptides were synthesized for subsequent screening (Supplementary Tables 2, 3 and Supplementary Data 1). Notably, most synthetic peptides showed highly similar fragmentation patterns and retention times to the discovery data, supporting the original identifications (Supplementary Data 1).

**CD8+ T-cells in IAV/HLA-A24 mice are biased toward polymerase-derived peptides**. To determine the immunogenicity of novel IAV- and IBV-derived peptides during primary and secondary influenza virus infection in vivo, we utilized HLA-A24-expressing transgenic (HHD-A24) mice[42]. These mice are not confounded by prior exposure to infections nor co-expression of

**Fig. 2 LC–MS/MS analyses of immunopeptidome from HLA-A*24:02+ influenza-infected cells. a** Length distribution of human proteome-derived HLA-I ligands of C1R.A24 isolated using the pan HLA-I antibody W6/32. Numbers of non-redundant sequences of ≤20 amino acids identified at a 5% FDR across the 12 experiments in which HLA class I was isolated from C1R.A24, filtered for peptides identified in HLA class II isolations at a 5% FDR. **b** Sequence logo derived from human 9mer peptides in **a** using Seq2logo2.0[75]. **c** Peptide length distribution of IAV (HKx31) and IBV (B/Malaysia) derived peptides identified as likely HLA-A*24:02 ligands in this study (no FDR cut-off applied) as shown in Supplementary Data 1. **d**, **e** Distribution of IAV- (**d**) and IBV-derived (**e**) A*24:02 ligands from **c** across the viral proteomes, including potential identifications from alternate reading frames (PB1 + 2, PB1 + 3, PB1-1, HA + 2, HA + 3 and NA + 3). **f**, **g** Identification of specific ligands derived from IAV (**f**) and IBV (**g**) in isolations performed at 2, 4, 8, 12, and 16 h (16 h IAV only) post infection. Colored squares represent the identification of each peptide derived from viral proteins as indicated in legend from **d**, **e**. In all panels, n = number of peptides.

other competing major histocompatibility complex I (MHC-I) molecules, and thus provide a valuable tool for screening peptide antigens in vivo. For primary IAV analyses, 6–10-week-old mice were intranasally (i.n.) infected with 200 pfu of H3N2 HKx31 virus, while secondary experiments were performed by an i.n. heterologous challenge with 200 pfu H1N1 PR8 6–8 weeks after the primary HKx31 infection. At day (d) 10 post-infection, spleen and bronchoalveolar lavage (BAL, combined from 4–5 mice) were stimulated with 54 peptides; including 48 novel peptides identified by mass spectrometry and the previously published $PB1_{496–505}$, $M1_{99–109}$, $M1_{108–117}$, $M1_{239–248}$, $NP_{39–47}$ & $NP_{419–429}$ peptides (Supplementary Table 2). The responses were detected by 5-h ex vivo peptide stimulation and measurement of IFN-γ production by ICS (Supplementary Fig. 2). Our data revealed that A24/IAV-specific CD8+ T-cell responses were immunodominant (>5% IFN-γ+ of CD8+ T-cells) towards 2 PB1- and 3 PB2-derived peptides: $PB1_{216–224}$ (mean of 10.2% IFNγ+ of CD8+ cells in spleen, 16.5% in BAL), the published $PB1_{498–505}$ (14.8% spleen, 23.3% BAL), $PB2_{549–557}$ (10.1% spleen, 24.9% BAL), $PB2_{549–559}$ (11.1% spleen, 24.4% BAL) and $PB2_{463–471}$ (6% spleen, 9.3% BAL) (Fig. 3a). Two additional subdominant (1% < IFN-γ+ of CD8+ T-cells < 5%) CD8+ T-cell responses were directed towards $PB2_{322–320}$ (3.9% spleen, 7.7% BAL) and previously published $NP_{39–47}$ (2.1% spleen, 7.7% BAL), while marginal CD8+ T-cell responses were detected, mainly in BAL, towards 6 other peptides ($HA_{248–259}$, $PA_{47–60}$, $PA_{130–138}$, $PB1_{2–10}$, $PB1_{500–505}$, $PB2_{703–710}$). Several key determinants can orchestrate the main target and therefore the immunodominance of CD8+ T cells. These include the affinity of the peptide to the HLA I molecule, the affinity of the CD8+ T cells for an epitope, as well as the number of naive epitope-specific CD8+ T cells (as reviewed in ref. [43]).

During the acute phase (d8) of secondary challenge, CD8+ T-cell responses were similarly directed at 6 immunogenic peptides found in the primary infection ($PB1_{216–224}$, $PB1_{498–505}$, $PB2_{322–330}$, $PB2_{463–471}$, $PB2_{549–557}$ and $PB2_{549–559}$) (Fig. 3b). CD8+ T-cell responses towards the marginal epitopes observed in the primary infection were no longer detected. While the reason for the loss of $HA_{248–259}$ could be explained by sequence variation between HKx31 and PR8 (IYWTIVKPGDVL vs. **Y**YWT**L**VKPGD**T**I) all internal proteins are shared between both viruses. Analysis of influenza-specific CD8+ T-cell numbers showed significant reductions in epitope-specific CD8+ T-cells for 9 out of 11 specificities ($NP_{39–47}$, $PA_{130–138}$, $PB1_{2–10}$, $PB1_{216–224}$, $PB1_{496–505}$, $PB1_{498–505}$, $PB2_{322–330}$, $PB2_{703–710}$ and $PB2_{549–559}$) following secondary infection (Supplementary Fig. 3). Changes in immunodominance after heterologous influenza virus infection have been studied previously, especially in B6 mice. Immunodominance hierarchy and cross-protective capacity of epitope-specific CD8+ T cells depend greatly on the first influenza encounter and presentation of viral epitopes by different cell types (reviewed in ref. [44]).

Thus, our in vivo screening in HHD-A24 mice identified 6 IAV-derived immunogenic peptides during primary and secondary IAV infection, with prominent CD8+ T-cell responses being heavily biased towards PB1- and PB2-derived peptides

(Supplementary Table 2). This is of key importance as the current T-cell vaccines in clinical trials focus mainly on internal proteins like NP and M1[45–49], which may be poorly immunogenic in HLA-A24-expressing individuals at risk of severe influenza disease.

**Prominent A24/CD8+ T-cell specificities in IBV infection.** While the CD8+ T-cell responses towards IBV have been studied in detail for HLA-A*02:01-expressing individuals[24], there remains a lack of known CD8+ T-cell epitopes for other HLAs. Here, we determined the immunogenicity of newly identified IBV-derived peptides using primary and secondary infection in HHD-A24 mice. HHD-A24 mice were i.n. infected with 200 pfu B/Malaysia/2506/2004. On d10, spleen and BAL were restimulated with individual peptides (Supplementary Table 3) to measure the corresponding CD8+ T-cell responses. Novel immunodominant IFN-γ-producing A24/CD8+ T-cell specificities were found against a range of IBV peptides including $NP_{164–173}$ (mean of 13.1% in the spleen, 31.15% in BAL), the shorter $NP_{165–173}$ (12.6% spleen, 38.4% BAL) and $NA_{32–40}$ (5.2% spleen, 17.3% BAL), while subdominant A24/CD8+ T-cell responses were detected towards an additional four IBV-derived peptides ($NA_{213–221}$, $NP_{392–400}$, $PB2_{245–253}$, $PB2_{550–558}$) (Fig. 3c).

To define IBV-specific CD8+ T-cell responses following the secondary challenge, B/Malaysia/2506/2004-infected mice were challenged i.n. after 6–8 weeks with 400 pfu B/Phuket/3073/2013. At d8 after challenge, IBV-specific CD8+ T-cell responses resembled those after primary IBV infection (Fig. 3d), but were focussed more towards the immunodominant epitopes, similar to IAV-specific A24/CD8+ T-cell responses following secondary challenge. Although the B/Phuket virus differs in one amino acid in the $NP_{164–173}$/$NP_{165–173}$ and $NA_{213–222}$ restimulating with the B/Malaysia variants showed cross-reactivity of CD8+ T-cells between both variants. All other immunogenic epitopes were conserved between both strains. In contrast to IAV infection, the total number of epitope-specific CD8+ T-cells for all immunogenic epitopes after secondary challenge remained comparable in the spleen (Supplementary Fig. 3). Thus, our data identified prominent A24/CD8+ T-cell responses directed toward IBV during primary and secondary influenza virus infection in HHD-A24 mice (Supplementary Table 3).

**IAV/CD8+ T-cells in HLA-A24+ Indigenous and non-Indigenous individuals.** Having identified prominent IAV-derived CD8+ T-cell epitopes towards the primary and secondary infection in HHD-A24 mice, it was of key importance to define immunodominant CD8+ T-cell sets in HLA-A24-expressing Indigenous and non-Indigenous individuals. For humans, different influenza strains of the same subtype are often co-circulating and mutating to create different variants of the same epitope region. In addition to our identified panel of HLA-A24-binding peptides, we searched the Influenza research database (https://www.ncbi.nlm.nih.gov/genomes/FLU/Database/nph-select.cgi?go=database) to

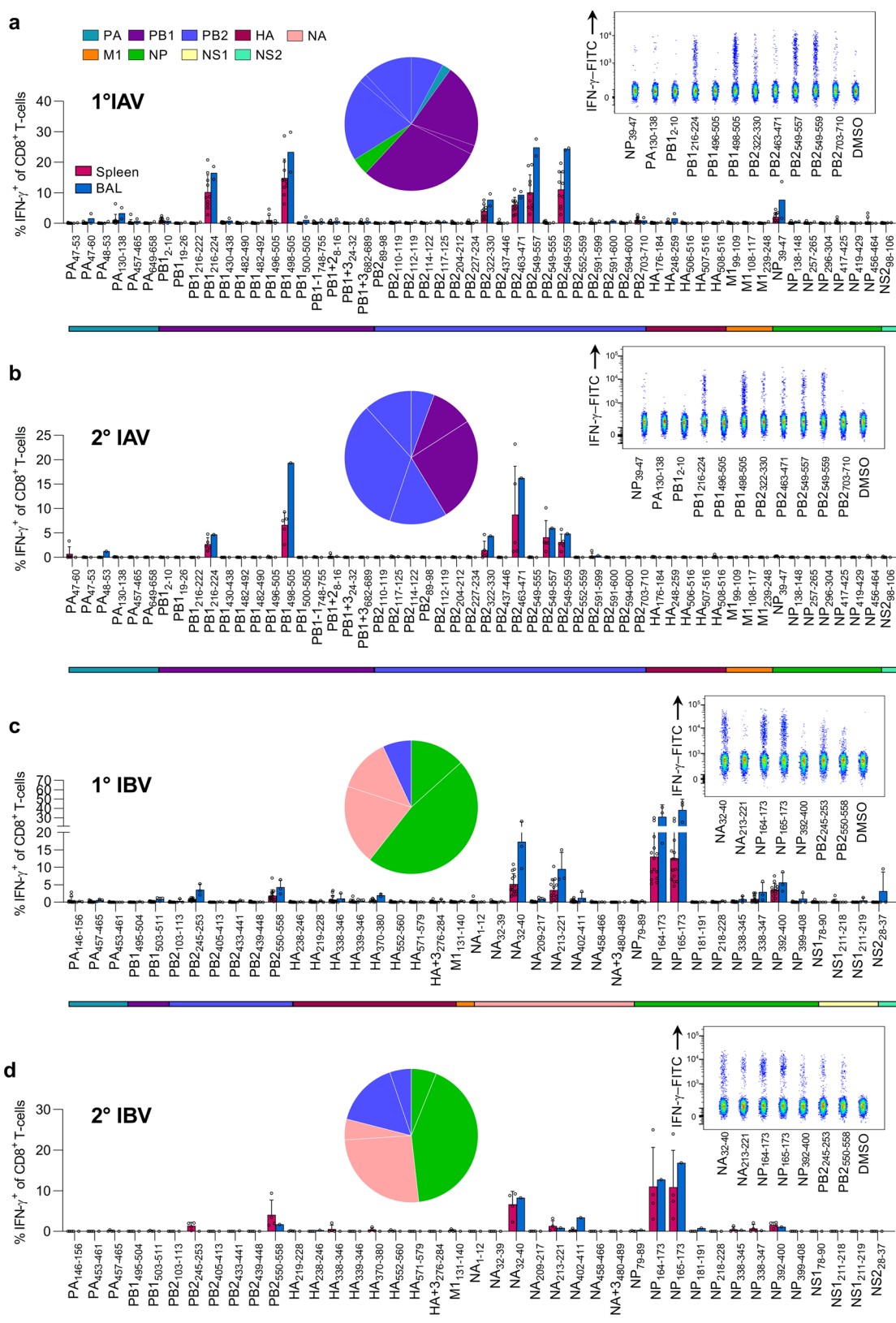

include the most frequent virus strains circulating in South-East Asia and Australia and identified naturally occurring variants of our HLA-A24 binding peptides utilizing the "Identify short peptides in proteins" analysis from the Influenza research database (fludb.org) ($n = 61–3877$ sequences dependent on protein) to include in epitope mapping (Supplementary Table 4). We

probed memory CD8[+] T-cell populations, by firstly stimulating PBMCs from healthy Indigenous HLA-A24-expressing donors with 5 IAV peptide pools for 13 days. Each pool contained 7–13 peptides (Supplementary Table 2), of which each variant (Supplementary Table 4) was always included in the same pool as the wildtype peptide identified in the immunopeptidome studies. We

**Fig. 3 Screening of immunogenic IAV and IBV peptides in HHD-A24 mice. a, c** Frequencies of IFN-$\gamma^+$ CD8$^+$ T-cells in the spleen (individual mice) and BAL (pooled mice) after primary infection with either IAV (100 pfu/30 μL i.n. A/HKx31) or IBV (200 pfu/30 μL i.n. B/Malaysia/2506/04), respectively, from 6 to 10 week-old HLA-A24-expressing mice. Cells from spleen and BAL were isolated 10 days post-infection and restimulated with individual peptides (**a**, $n = 10$ mice from 2 independent experiments, except PB2$_{591-600}$, PB2$_{117-125}$ 5 mice from 1 experiment) (**c**, $n = 14$ mice from 3 experiments; except NA$_{209-217}$ $n = 13$ from 3 independent experiments; NS$_{28-37}$, NP$_{399-408}$, NS$_{178-190}$, NA$_{1-12}$, NP$_{338-345}$, NP$_{181-191}$, $n = 12$ mice from 3 experiments; HA$_{370-380}$, NA + 3$_{480-489}$ $n = 10$ mice 2 experiments; NP$_{218-228}$, PA$_{146-156}$, NS1$_{211219}$, NA$_{32-39}$, HA$_{339-346}$, NA$_{458-466}$, NS1$_{211-218}$ $n = 9$ mice from 3 experiments). **b, d** IFN-$\gamma^+$ frequencies following a secondary heterologous challenge with IAV (200 pfu/30 μL of A/PR8) and IBV (400 pfu/30 μL of B/Phuket/3073/2013), 6 weeks after primary infection, respectively. Cells were isolated 8 days post-infection for ICS assay (**b**, $n = 5$ mice from 1 experiment) (**d**, $n = 4$ mice from 1 experiment). **a–d** Bars indicate mean + SD for spleen and BAL for **c** only. Pie charts depict the protein origin of immunogenic peptides and the average contribution to the total IFN-$\gamma$ response in the spleen for each primary and secondary IAV/IBV responses. Colored bars below the bar chart indicate the protein origin of characterized peptides. Representative concatenated FACS plots from splenocytes of immunodominant and subdominant responses compared to DMSO background control are shown in the top-right panels.

observed CD8$^+$ T-cell responses towards pools 1 and 2 via an IFN-$\gamma$/TNF ICS assay (Fig. 4a and Supplementary Fig. 4), and subsequently, cell cultures from those pools were restimulated with individual peptides (+variants) to map the immunogenic epitopes. In 5/5 Indigenous donors tested, CD8$^+$ T-cell responses were dominated by PB1$_{498-505}$ (median of 0.93% IFN-$\gamma$ of CD8$^+$ T-cells; also immunodominant in mice Fig. 3a, b), its longer 10mer version PB1$_{496-505}$ (median of 0.64%; detected marginally after 1° IAV infection in mice), PA$_{649-658}$ (median of 0.73%; not detected in mice) and NP$_{39-47}$ (median 0.16%, immunogenic following 1° IAV infection in mice). Another 6 subdominant HLA-A24-restricted epitopes were identified in a select number of donors (PB2$_{110-119}$ in 4/5 donors, NS2$_{98-106}$ in 3/5, PB1$_{216-224}$ in 3/5, PB2$_{549-557}$ in 1/5, HA$_{176-184}$ in 3/5 and PB2$_{703-710}$ in 3/5), of which PB1$_{216-224}$, PB2$_{549-557}$ and PB2$_{703-710}$ were also detected in HHD-A24 mice. Interestingly, only 2/4 immunodominant epitopes observed in the Indigenous donors elicited comparably robust responses in 5 non-Indigenous donors screened (published epitopes: PB1$_{498-505}$ median 0.93% vs 1.1%, PB1$_{496-505}$ median 0.64% vs 0.8%), while the other two epitopes, PA$_{649-658}$ and NP$_{39-47}$ were poorly immunogenic in non-Indigenous donors who instead responded well to the PB2$_{549-557}$ epitope, absent in 4/5 Indigenous donors. Such differential epitope preference and immunodominance hierarchies between Indigenous and non-Indigenous donors is perhaps influenced by different HLA co-expression backgrounds or infection history.

**A24/CD8$^+$ T-cells toward IBV in Indigenous and non-Indigenous donors.** Following identification of novel IBV-derived peptides in HHD-A24 mice, we defined CD8$^+$ T-cell responses towards IBV peptide pools, comprising 41 peptides identified by mass spectrometry (Supplementary Table 3) and an additional 14 variants (Supplementary Table 4), in HLA-A24-expressing Indigenous and non-Indigenous donors. In accordance with the HHD-A24 mouse data, we found broad A24/CD8$^+$ T-cell responses directed towards pool 10, which mapped to 6 major immunogenic epitopes spanning 5 different proteins (NP$_{164-173}$/NP$_{165-173}$, NA$_{32-40}$, PB2$_{550-558}$, PA$_{457-465}$, HA$_{552-560}$ and PB1$_{503-511}$) (Fig. 4b and Table 3). CD8$^+$ T-cell responses to these peptides were found in 7/9, 8/9, 6/9, 6/9, 5/9, and 5/9 of Indigenous and 5/5, 5/5, 4/5, 5/5, and 4/5 of non-Indigenous donors, respectively, with comparable IFN$\gamma^+$CD8$^+$ T-cell frequencies between Indigenous and non-Indigenous donors. The diversity of CD8$^+$ T cell epitopes was comparable between Indigenous (LIFT) and non-Indigenous (non-LIFT) PBMCs.

As our experiments examined the immunogenicity of A24/CD8$^+$ T-cells following in vitro peptide-driven expansion, we further verified these novel IBV epitopes in non-Indigenous donors by incubating HLA-A24$^+$ PBMCs with IBV-infected (B/Malaysia/2505/2004) C1R.A24 target-cells for 15 days before

dissecting individual peptide (+variant) responses from IBV pool 10. This confirmed the immunogenicity of at least 3 epitopes previously detected by peptide-driven expansions (NP$_{164-173}$/NP$_{165-173}$, NA$_{32-40}$, and PB2$_{550-558}$) (Fig. 4c). Interestingly, the same epitopes were observed in HHD-A24 mice after IBV infection (Fig. 3c, d).

Collectively, out of 41 HLA-A24-binding IBV epitopes newly identified by immunopeptidomics, we confirmed a total of 9 (22%) immunogenic epitopes after screening HLA-A24-expressing mice and humans. Of these, 3 were found in both humans and mice (NP$_{164-173}$/NP$_{165-173}$, NA$_{32-40}$, PB2$_{550-558}$), 3 were only found in humans (HA$_{552-560}$, PA$_{457-465}$ and PB1$_{503-511}$), and the remaining 3 were only found in mice (PB2$_{245-253}$, NA$_{213-221}$ and NP$_{392-400}$).

**A24/CD8$^+$ T-cell cross-reactivity between IAV and IBV to PB2 epitopes.** The B/Malaysia/2505/2004 virus used here is from the Victoria (Vic) lineage, however, there is another IBV lineage that commonly co-circulates and infects humans called the Yamagata (Yam) lineage. To determine the potential for HLA-A24/CD8$^+$ T-cell cross-reactivity across both IBV lineages, B/Malaysia/2505/2004 (Vic) virus-expanded CD8$^+$ T-cells were restimulated with IBV peptide variants from the alternate Yam lineage. Cross-reactivity was demonstrated towards the NP$_{165-173}$ epitope (YFSPIRVTF variant 1 (Vic only)), in which B/Malaysia/2505/2004-expanded CD8$^+$ T-cells responded towards the Yam variant (YFSPIR**I**TF v2 (Yam only)) following ICS assay (Fig. 4d). Likewise, cross-reactivity between Vic lineage and a variant found in the Yam lineage was also observed with the PB2$_{550-558}$ variants (TYQWVLKNL (variant 1, both Vic and Yam); TYQWV**M**KNL (variant 2, Yam only)) in the virus-expansion system, but 3/4 donors did only respond to v1 after peptide expansion (Fig. 4d).

Koutsakos et al. have previously reported cross-reactivity towards IAV and IBV (as well as ICV) in the HLA-A2 model with a single epitope sequence[24]. Here, none of the identified HLA-A24 IAV and IBV epitopes showed 100% sequence identity between strains. Instead, we identified a potential HLA-A24-restricted IAV/IBV cross-reactive candidate, the immunogenic IAV PB2$_{549-557}$ TYQW**IIRNW** epitope. This epitope shares 55% amino acid identity with the cross-reactive IBV PB2$_{550-558}$ variants. Indeed, virus-expansion with IAV/HKx31-infected C1R.A24 cells induced cognate IAV/PB2$_{549-557}^+$ (A/PB2$_{549-557}$) CD8$^+$ T-cell responses in 2/4 donors as well as cross-reactive responses to the IBV PB2$_{550-558}$v1 (B/PB2$_{550-558v1}$) variant and the B/PB2$_{550-558}$v2 variant (Fig. 4e). The superiority of IBV at inducing cross-reactive CD8$^+$ T-cells in contrast to IAV was validated with the reverse experiment, where virus-expansion with B/Malaysia/2506/2004-infected C1R.A24 cells induced highly robust CD8$^+$ T-cell responses in 4/4 donors, capable of cross-recognizing all three different epitopes (Fig. 4e).

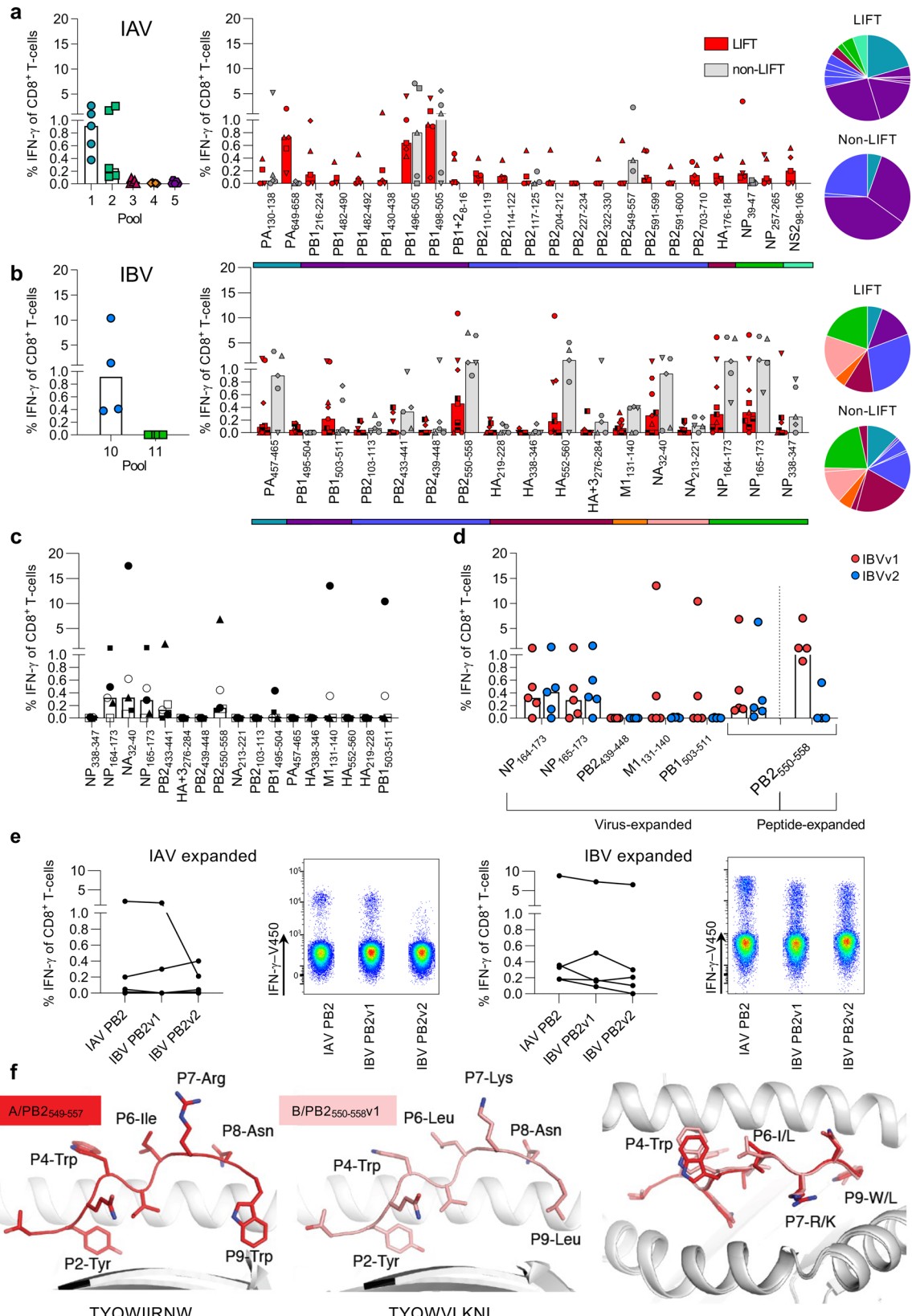

As we observed T-cell cross-reactivity between the A/PB2$_{549-557}$ and B/PB2$_{550-558}$ (Fig. 4e) as well as T-cell reactivity in transgenic mice for the overlapping peptide A/PB2$_{549-559}$, we next determined the impact of those variations within the PB2-derived peptides. We solved the structures of A/PB2$_{549-557}$, the overlapping A/PB2$_{549-559}$, and B/PB2$_{550-558}$ peptides in complex

with the HLA-A24 molecules at a resolution of 2.90, 2.95, and 2.16 Å, respectively (Supplementary Table 5) with clear electron density for each peptide (Supplementary Fig. 5a–f).

The 9mer A/PB2$_{549-557}$ peptide adopted a canonical extended conformation within the cleft of HLA-A24, with anchor residues at P2-Tyr and P9-Trp, and a secondary anchor residue at P5-Ile.

**Fig. 4 In vitro screening for immunogenicity of IAV and IBV epitopes in human PBMC. a, b** Frequencies of IFN-γ$^+$CD8$^+$ T-cells after PBMCs were expanded with IAV (**a**) or IBV (**b**) peptide pools for 13 days and restimulated with C1R.A24 cells pulsed with the corresponding peptide pools (left panel; IAV $n = 5$ biologically independent samples, IBV $n = 4$ biologically independent samples). Right panels show dissection of each peptide from IAV pools 1 and 2 (LIFT $n = 5$, non-LIFT $n = 3$–5) (**a**) or IBV pool 10 (LIFT $n = 9$ biologically independent samples, non-LIFT $n = 5$ biologically independent samples) (**b**) using single peptide-pulsed C1R.A24 cells after 15 days of peptide-pool T-cell expansions in LIFT and non-LIFT. **a, b** Pie charts depict protein origin of immunogenic peptides and the median contribution to the total IFN-γ response for Indigenous (LIFT) and non-Indigenous (non-LIFT) donors. Colored bars below the bar chart indicate the protein origin of characterized peptides. **c, d** Individual IBV peptide IFN-γ$^+$CD8$^+$ T-cell responses in non-LIFT donors following expansion with B/Malaysia/2506/04-infected C1R.A24 cells for 15 days and ICS. **c** For the ICS, peptides derived from B/Malaysia/2506/04 or **d** comparison between autologous (v1) and the alternative circulating variant (v2) was used. Symbols indicate individual donors screened ($n = 5$ biologically independent samples). **d** IBV PB2$_{550-558}$ v1 and v2 responses after virus expansion versus IBV PB2$_{550-558}$ v1 peptide expansion are shown on the right. **a–d** Bars indicate median. **e** Cross-reactive PB2 responses of IAV PB2, IBV PB2v1, and IBV PB2v2 peptides following PBMC expansion with IAV (A/HKx31) (left panels; $n = 4$ biologically independent samples) or IBV (B/Malaysia/2506/04) infected (right panels; $n = 5$ biologically independent samples) C1R.A24 cells from 5 non-LIFT donors with representative concatenated FACS plots. **f** Crystal structures of HLA-A*24:02 presenting the PB2 peptide variants IAV PB2 (red, left) and IBV PB2v1 (pink, middle), and an overlay of both peptides (right).

| Table 3 Human immunodominant A24 epitopes. | | | |
|---|---|---|---|
| **Peptide** | **Sequence** | **Virus origin** | **Source** |
| PA$_{649-658}$ | LYASPQLEGF | IAV | Previous identified/Mass spectrometry |
| PB1$_{496-505}$ | FYRYGFVANF | IAV | Previously identified |
| PB1$_{498-505}$ | RYGFVANF | IAV | Previous identified/Mass spectrometry |
| PB2$_{549-557}$ | TYQWIIRNW | IAV | Previous identified/Mass spectrometry |
| PA$_{457-465}$ | KYVLFHTSL | IBV | Mass spectrometry |
| PB1$_{503-511}$ | NFAMELPSF | IBV | Mass spectrometry |
| PB2$_{550-558}$ | TYQWVLKNL | IBV | Mass spectrometry |
| HA$_{552-560}$ | YYSTAASSL | IBV | Mass spectrometry |
| NA$_{32-40}$ | LYSDILLKF | IBV | Mass spectrometry |
| NP$_{165-173}$ | YFSPIRVTF | IBV | Mass spectrometry |

Solvent exposed residues were at P4-Trp, P6-Ile, P7-Arg, and P8-Asn, representing a large surface accessible for TCR interaction. The P9-Trp of the peptide formed a network of interactions with HLA-A24 tyrosine residues at positions 116, 118, and 123 as well as the Leu95 (Supplementary Fig. 3a–c), likely assisting with stabilizing the complex reflected in the higher stability observed for the HLA-A24-A/PB2$_{549-557}$ complex than with other peptides (Supplementary Table 5).

The B/PB2$_{550-558}$ peptide differed from the A/PB2$_{549-557}$ peptide at positions 5 (Ile to Val), 6 (Ile to Leu), 7 (Arg to Lys), and 9 (Trp to Leu) (Fig. 4f). Both peptides shared the same anchor residue at P2-Tyr and similar solvent-exposed residues (except for P7-Lys) but differed at PΩ (P9). As Leu possessed a shorter side chain than Tyr at PΩ, the IBV peptide was not buried as deeply into the F pocket, which may explain the lower stability observed for the B/PB2$_{550-558}$ peptide ($T_m$ of 57 °C) compared to A/PB2$_{549-557}$ ($T_m$ of 62 °C) (Supplementary Table 5). Structural overlay of HLA-A24 presenting the A/PB2$_{549-557}$ and B/PB2$_{550-558}$ peptides showed that the antigen-binding cleft and both peptides adopted a similar conformation with an average root mean square deviation (r.m.s.d.) of 0.31 and 0.37 Å, respectively (for Cα atoms) (Fig. 4f), consistent with T-cell cross-reactivity observed towards these two peptides.

Although the 11mer A/PB2$_{549-559}$ generated similar responses to the 9mer A/PB2$_{549-557}$ in HHD-A24 mice (Fig. 3a, b), it was not immunogenic in peptide-pool screening in humans (Supplementary Table 2 Pool 4) (Fig. 4a) as perhaps the minimal 9mer epitope was not exposed for T-cell recognition, due to the two additional C-terminal residues (P10-Glu and P11-Thr) (Supplementary Fig. 6a, b). Similar to the 9mer peptide conformation P2-

Tyr and P9-Trp of the 11mer PB2$_{549-559}$ act as primary anchor residues buried in the HLA-A24 antigen-binding cleft with the structural overlay of the peptides showing a r.m.s.d. of 0.48 Å (Supplementary Fig. 6a–c). Strikingly, the extra P10-Glu and P11-Thr residues of the 11mer extended outside the antigen-binding cleft, creating an unusual conformation that disturbed the interaction between the peptide and the HLA-A24 Lys146 at the C-terminal of the cleft. The Lys146 residue is a conserved residue in HLA molecules that helps stabilize the pHLA complex[50]. In the 9mer PB2$_{549-557}$ peptide, Lys146 interacts with the carboxylic group of the PΩ residue (Supplementary Fig. 6d), however, this interaction is lost in the 11mer due to the presence of the extra two residues P10-Glu and P11-Thr (Supplementary Fig. 6e), thereby likely decreasing the pHLA stability compared to the shorter A/PB2$_{549-557}$ peptide (Supplementary Table 5). Thus, the bulged conformation of the extra residues in the A/PB2$_{549-559}$ may represent a challenge for TCRs interacting with the C-terminal end of the peptide. In compliance with the in vitro data, the structural data support the potential cross-reactivity of CD8$^+$ T-cells between the A/PB2$_{549-557}$ and the B/PB2$_{550-558}$, verifying our previous findings of broad CD8$^+$ T-cell immunity against influenza virus infections.

**HLA-A24 presents overlapping NP$_{164-173}$ and NP$_{165-173}$ in different conformations.** We observed robust comparable mouse (Fig. 3c, d) and human (Fig. 4b) responses to the overlapping IBV NP$_{164-173}$ and NP$_{165-173}$ peptides. This suggested that the 9mer NP$_{165-173}$ peptide would be the minimal epitope and both peptides might present a similar core confirmation recognized by T-cells. To determine this possibility, we solved the structures of the overlapping B/NP$_{164-173}$ and B/NP$_{165-173}$ peptides in complex with HLA-A24 at a resolution of 2.75 and 1.51 Å, respectively (Supplementary Figs. 5g–j, 7 and Supplementary Table 5). The B/NP$_{164-173}$ and B/NP$_{165-173}$ peptides adopted a canonical extended conformation in the cleft of HLA-A24 molecule (Supplementary Fig. 7). P2-Phe and P9-Phe anchor residues of the 9mer B/NP$_{165-173}$ were buried deep inside the hydrophobic B and F pockets of the HLA (Supplementary Fig. 7a). Three residues were exposed to the solvent for possible TCR recognition (P1-Tyr, P6-Arg, P8-Thr). The P5-Ile and P7-Val of this 9mer peptide were partially buried (Supplementary Fig. 7e).

Structural overlay of 9mer and 10mer B/NP peptides were different due to the 10mer's extra residue at the N-terminus (r.m.s.d. of 1.36 Å), which shifted the anchor residues (Supplementary Fig. 7b, c). The substitution of P2-Tyr (NP$_{164-173}$) for P2-Phe (NP$_{165-173}$) occurred without major structural rearrangement, as both residues were large aromatic residues filling the B pocket (Supplementary Fig. 7f). However, the additional residue changed the secondary anchor residue at P3 from a small P3-Ser

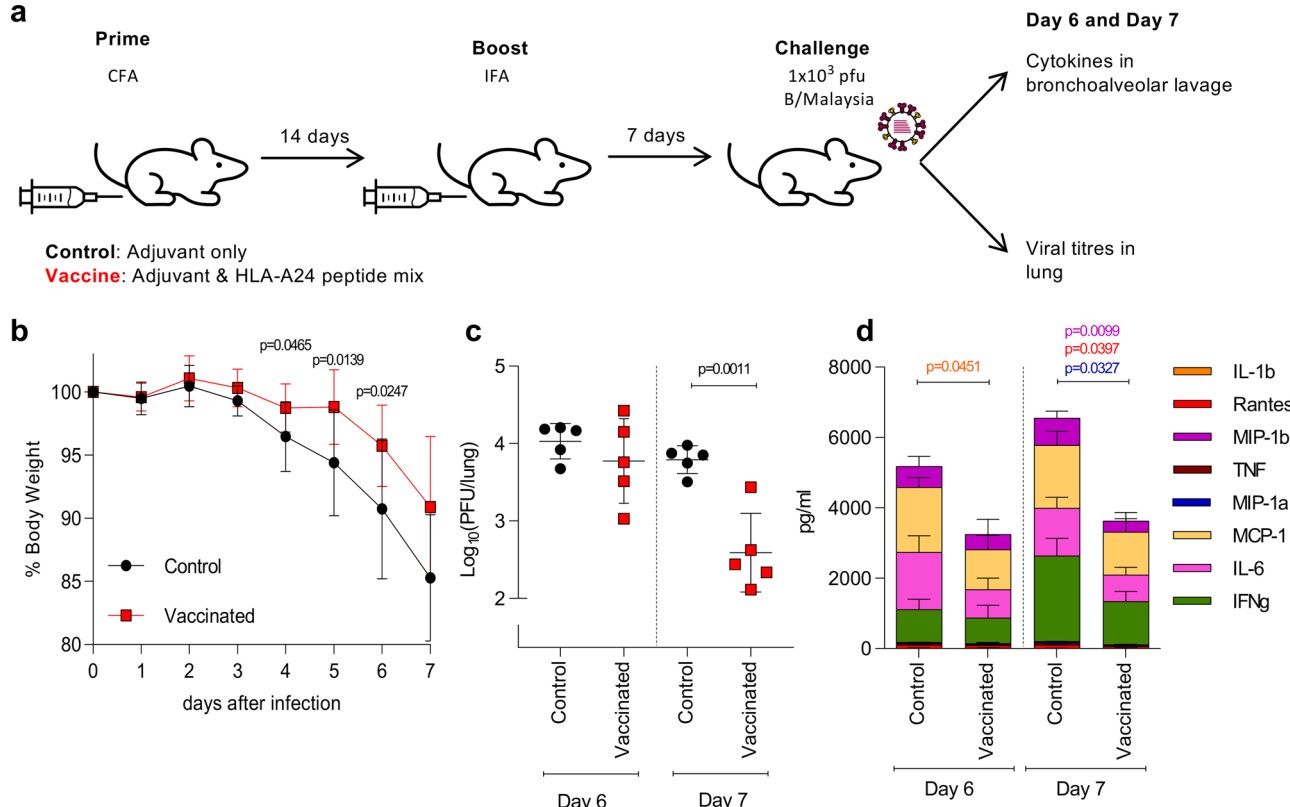

**Fig. 5 Protection of HDD.24 mice by vaccination with immunogenic A24 peptides. a** Mice were vaccinated with 3 immunogenic IBV peptides (NP$_{164}$, NP$_{392}$, NA$_{32}$) in adjuvant CFA or CFA alone (control). 14 days post-priming, immune responses were boosted with peptides/IFA or IFA alone. 7 days after the boost, mice were challenged with a high dose of $1 \times 10^3$ pfu B/Malaysia. Bronchoalveolar lavage and lungs were collected on days 6 and 7 post infection. **b** % bodyweight of mice after virus challenge at different days after infection. **c** Virus titer from homogenized lungs on days 6 and 7 post challenge. **d** Cytokine levels in BAL on day 6 and 7. Colored stars indicate significant differences between control and peptide-vaccinated groups. **b–d** Statistical significance was calculated using an unpaired two-tailed t-test with *p < 0.05 (n = 5 animals per group, 1 experiment). Mean values are shown with ±SD (**b, c**) or +SD (**d**).

(NP$_{165–173}$) to a large P3-Phe (NP$_{164–173}$) (Supplementary Fig. 7g). The larger P3-Phe might stabilize the B pocket of the HLA-A24 better than the small P3-Ser, and therefore could explain the 7 °C higher Tm observed for the NP$_{164–173}$ than the NP$_{165–173}$ in complex with HLA-A24 (Supplementary Table 5), which could also reflect the immunogenicity of this peptide. The largest structural difference between the two peptides was observed at the center of the peptide (P6/7-Arg) with a maximum displacement of 3.9 Å for the Cα atom (Supplementary Fig. 7d). The P7-Arg of the NP$_{164–173}$ peptide sat higher than the backbone of the NP$_{165–173}$ peptide (Supplementary Fig. 7c) and was a prominent feature for potential TCR interaction of the 10mer peptide, contrasting with the hydrophobic nature of the 9mer NP$_{165–173}$ peptide. Thus, the structures of HLA-A24 presenting the two NP peptides showed that, despite being overlapping peptides that differ only by one residue, the NP$_{164–173}$ and NP$_{165–173}$ peptides adopt different structural conformations. As a result, both peptides exposed different residues to the solvent, and hence would most likely be recognized by different TCRαβ repertoires.

**Protective capacity of novel HLA-A24 IBV-epitope in HHD-A24 mice.** To determine the protective capacity of the novel CD8$^+$ T cell epitopes in HHD-A24 mice, we performed a proof of principle experiment and vaccinated mice with 3 immunogenic IBV peptides (NP$_{164}$, NP$_{392}$, NA$_{32}$) using a well-established prime/boost approach[24], then infected mice i.n. with $1 \times 10^3$ pfu B/Malaysia (Fig. 5a). Vaccination with HLA-A24-restricted

peptides resulted in significant protection against IBV. This was shown by decreased disease severity on d4, d5, and d6 after IBV infection as measured by the bodyweight loss (Fig. 5b; $p < 0.05$, unpaired two-tailed t-test) as well as a significant ~89% reduction in viral titers in the lung on d7 after IBV infection when compared to the mock-immunized group ($p = 0.001$, unpaired two-tailed t-test) (Fig. 5c). Additionally, there was a significant decrease ($p < 0.05$, unpaired two-tailed t-test) in the levels of inflammatory cytokines (MIP-1β, MIP-1a, RANTES) in d7 BAL of peptide-vaccinated mice in comparison to the mock-immunized animals (Fig. 5d). Thus, CD8$^+$ T cells directed at our novel HLA-A24-restricted IBV-specific epitopes provide a substantial level of protection against influenza disease, as they can markedly decrease body weight loss, accelerate viral clearance and reduce the cytokine storm at the site of infection.

**A24/CD8$^+$ T-cells in influenza patient blood and healthy human tissues.** Having identified the prominent IAV and IBV CD8$^+$ T-cell specificities for Indigenous and non-Indigenous HLA-A24$^+$-individuals, we sought to determine whether CD8$^+$ T-cells specific for our newly identified epitopes were recruited and activated during acute influenza virus infection. We generated peptide/HLA-A24-tetramers to the most immunogenic IAV (A/PB1$_{498–505}$) and newly characterized IBV epitopes (NP$_{165–173}$ and NA$_{32–40}$). These reagents allowed direct ex vivo detection of IAV- and IBV-specific CD8$^+$ T-cells using tetramer-associated magnetic enrichment (TAME)[51] in both healthy and influenza-infected individuals (Fig. 6a, left panels and Supplementary

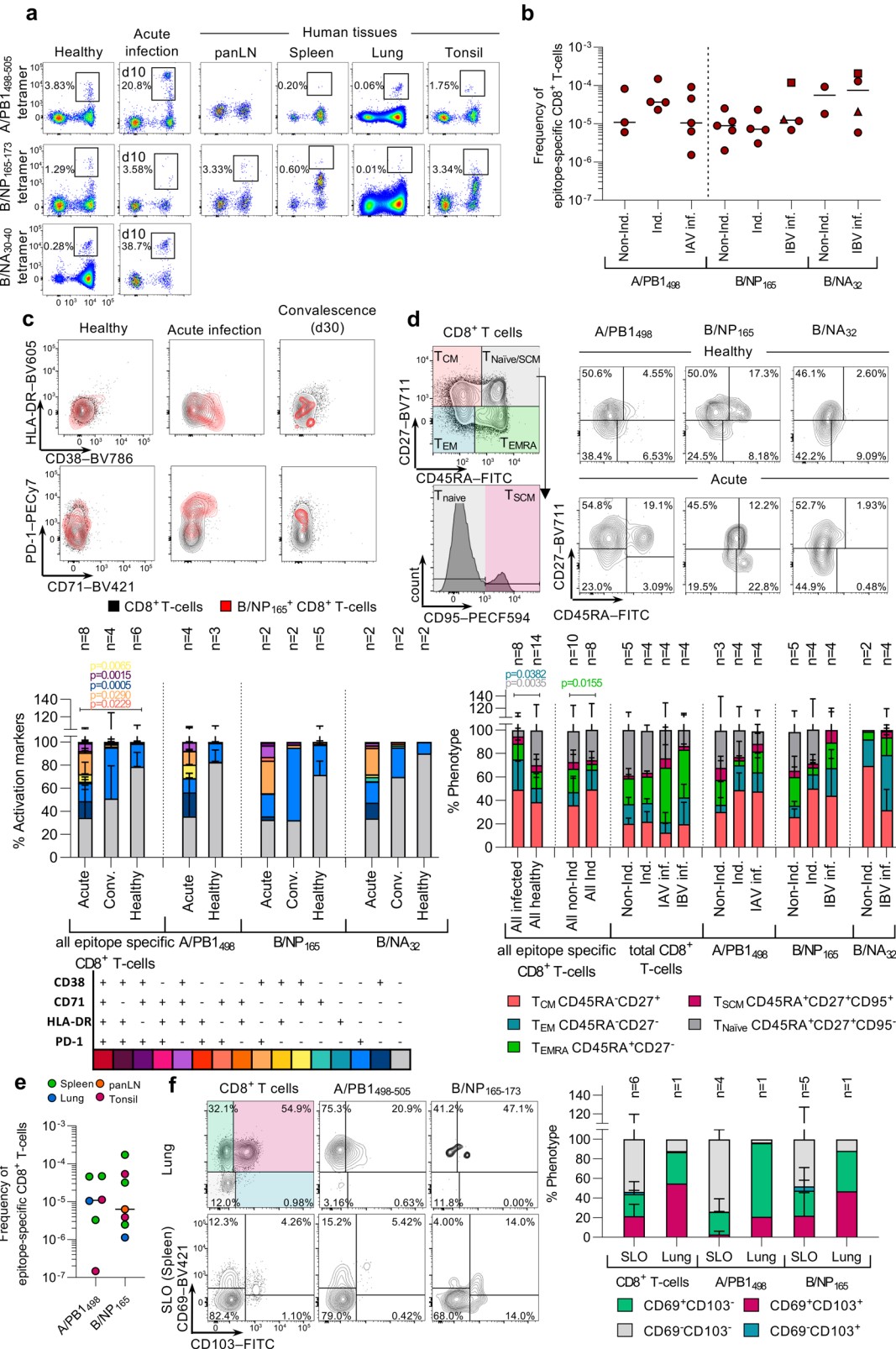

Fig. 8). In healthy non-Indigenous and Indigenous donors, ex vivo mean precursor frequencies for A/PB1$_{498-505}$$^+$ and B/NP$_{165-173}$$^+$ CD8$^+$ T-cells, were $4 \times 10^{-5}$ and $1 \times 10^{-5}$ of CD8$^+$ T-cells, respectively (Fig. 6b). Non-Indigenous B/NA$_{32}$$^+$ frequencies were $1.8-9 \times 10^{-5}$ of CD8$^+$ T-cells. All tetramer$^+$ frequencies fell within the range of previously published frequencies for memory IAV- or EBV-specific CD8$^+$ T-cells[51,52].

Interestingly, as per our analysis in HLA-A*02:01-positive influenza patients[24], the frequencies of A/PB1$_{498}$$^+$, B/NP$_{165}$$^+$ and B/NA$_{32}$$^+$ CD8$^+$ T-cells in blood of IAV- and IBV-infected patients during infection were comparable to that of memory CD8$^+$ T-cell frequencies (Fig. 6b) This, most likely reflects the accumulation of influenza-specific CD8$^+$ T-cells at the site of infection rather than in the peripheral blood. However, despite

**Fig. 6 Ex vivo IAV- and IBV-specific CD8$^+$ T-cells detected in HLA-A24+ PBMCs and tissues. a** Tetramer-positive cells of three different specificities (A/PB1$_{498-505}$, B/NP$_{165-173}$, B/NA$_{32-40}$) were enriched in human blood and tissues. Representative FACS plots of enriched fractions are shown for healthy donors, acute influenza-infected patients, and tissues from deceased organ donors. **b** Tetramer$^+$ precursor frequencies in the blood of healthy LIFT and non-LIFT, and influenza-infected donors where acute and convalescent time-points are shown together (IBV-inf). Samples from donor IBV2-inf were collected at acute square and convalescent timepoint triangle. **c** Activation status of epitope-specific CD8$^+$ T-cells in healthy, acute and convalescent donors. Statistical significance was calculated using a two-tailed Mann–Whitney test. **d** T-cell differentiation phenotype of epitope-specific cells in healthy LIFT and non-LIFT donors, and influenza-infected patients. **e** Tetramer$^+$ precursor frequencies of epitope-specific cells in human tissues. **f** CD103 and CD69 expression on epitope-specific cells in the lung compared to secondary lymphoid organs (SLO; CD8$^+$ T cells 2 spleens, 1 lymph node, 3 tonsils; A/PB1$_{498}$-specific cells 2 spleens, 2 tonsils; B/NP$_{165}$-specific cells 2 spleens, 1 lymph node, 2 tonsils). **b, e** Lines represent the median. **c, d, f** Bars represent mean + SD.

similar frequencies, their activation profiles varied greatly, as assessed directly ex vivo by expression of the activation markers CD71, HLA-DR, CD38, PD-1, and phenotypic markers CD27 and CD45RA (Fig. 6c, d). In patients acutely (within 10 days post disease onset) infected with IAV and IBV, tetramer$^+$CD8$^+$ T-cells displayed cells were expressing activation markers in significantly ($p < 0.05$, two-tailed Mann–Whitney test) higher frequencies, especially CD38$^+$PD-1$^+$ (18.7% vs. 0.8%), CD71$^+$PD-1$^+$ and CD38$^+$ (14.6% vs. 0.1%), CD38$^+$CD71$^+$PD-1$^+$ (6.7% vs 0.0%) and CD38$^+$CD71$^+$ (6.3% vs. 0.0%), in comparison to epitope-specific CD8$^+$ T-cells of healthy donors (Fig. 6c). In convalescent donors (30 days post disease onset), the dominant activation factor expressed in the epitope-specific CD8$^+$ T cells was PD-1 (44.2% in convalescent vs. 18.4% in healthy). Phenotypic analysis of CD45RA, CD27, and CD95 expression confirmed these results and showed higher frequencies of CD45RA$^-$CD27$^-$ effector memory-like T-cells in IAV- and IBV-infected patients (mean of 25.9% vs. 12.2% in healthy, $p = 0.04$, Mann–Whitney test), higher (but not significant) frequency of CD45RA$^-$CD27$^+$ central memory-like T-cells (49.1% vs. 38.5% in healthy, $p = 0.24$, Mann–Whitney test), and less naive T-cells (5.4% vs. 29.7% in healthy, $p = 0.004$, two-tailed Mann–Whitney test) compared to healthy individuals within the pooled IAV/IBV tetramer specificities (Fig. 6d). Comparing all healthy donors, we observed a lower frequency of CD45RA$^+$CD27$^-$ T$_{EMRA}$-like T-cells in epitope-specific CD8$^+$ T-cells from Indigenous- compared to non-Indigenous donors (5.0% vs 20.0% in non-Indigenous, $p < 0.05$, two-tailed Mann–Whitney test) (Fig. 6d). This difference was not observed in the total non-specific CD8$^+$ T-cell population.

Importantly, HLA-A24-restricted influenza-specific CD8$^+$ T-cells against A/PB1$_{498}$ and A/NP$_{165}$ were detected in multiple healthy human tissues directly ex vivo (Fig. 6a, right panels and Supplementary Fig. 9). A/PB1$_{498}$-specific CD8$^+$ T-cells were detected in the lung, spleen, and tonsil at frequencies ranging from $1.5 \times 10^{-7}$ to $4.5 \times 10^{-5}$ of total CD8$^+$ T-cells ($n = 6$ data points) but were not detected in the pancreatic lymph node (panLN) of one donor that had detectable A/PB1$_{498-505}$-specific CD8$^+$ T-cells in the spleen (Fig. 6e). B/NP$_{165}$-specific CD8$^+$ T-cells were found across all the tissues (lung, spleen, tonsil, panLN, $n = 7$ data points) at frequencies between $1.1 \times 10^{-6}$ and $1.7 \times 10^{-4}$. In human lung, IAV/IBV-specific CD8$^+$ T-cells had large populations of CD69$^-$CD103$^+$ and CD69$^+$CD103$^+$ tissue-resident memory (T$_{RM}$) T-cells (A/PB1$_{498-505}$: 75% and 21%, B/NP$_{165-173}$: 41% and 47% of tetramer-specific CD8$^+$ T-cells, respectively) (Fig. 6f). Secondary lymphoid organs (SLOs) were predominantly CD69$^-$CD103$^-$ circulating effector memory cells (range 23.8–85.7%).

Our findings demonstrate the presence of highly activated influenza-specific CD8$^+$ T-cells against the published A/PB1$_{498}$ epitope and the IBV epitopes identified here in HLA-A24$^+$ patients with acute influenza infection and memory pools

across different human tissues, highly relevant to the Indigenous population.

**Distinct HLA-A24 pMHC tetramer staining patterns reflect KIR3DL1 binding.** It was apparent from the tetramer-enrichment assays that some healthy donors contained large populations of HLA-A24-tetramer-binding CD8$^+$ T-cells prior to enrichment (up to 6% of CD8$^+$ T-cells) (Fig. 7a). This appeared to be donor-dependent but not entirely CD8$^+$ T-cell specificity-dependent. We found such oversized (0.32–6.73% in unenriched PBMCs) tetramer$^+$CD8$^+$ T-cell populations for A/PB1$_{498}$ in 10 out of 23 donors and in 14 out of 26 donors for B/NP$_{165}$ tetramers, but not for B/NA$_{32}$ (0/4 donors), which was further enriched with TAME (Fig. 7a, b). Ex vivo peptide stimulation and ICS performed with PBMCs from Non-LIFT 8 donor showed minimal IFNγ/TNF response (Supplementary Fig. 10), despite the high frequency of tetramer-positive CD8$^+$ T cells, indicating that these high frequency, low-affinity tetramer-binding ex vivo populations were not peptide-specific. Thus, it is important to note that our tetramer analyses in Fig. 5 excluded this oversized low-intensity staining tetramer-binding CD8$^+$ T-cell population. Such oversized tetramer-binding CD8$^+$ T-cell population could potentially be a unique HLA-A24-tetramer-binding phenomenon occurring in selected donors and hence potentially impair TCR-specific CD8$^+$ T-cell binding. Therefore, we sought to better understand HLA-A24-tetramer-binding in donors with conventional and largely oversized HLA-A24-tetramer CD8$^+$ T-cell populations.

Phenotypic analyses comparing tetramer-enriched fractions revealed that tetramer-binding CD8$^+$ T-cells of donors with oversized populations were predominantly of the CD45RA$^+$CD27$^-$ effector (T$_{EMRA}$) phenotype (mean 73.3 and 71.7% for PB1$_{498}$ and NP$_{165}$, respectively), while those from donors with conventional tetramer$^+$CD8$^+$ T-cells were predominantly T$_{CM}$ (mean 31.6 and 45.5%), T$_{EM}$ (10.3 and 8.3%) and T$_{Naive}$ (12.5 and 22.6%) in phenotype (Fig. 7c). To determine factors underlying this phenomenon, we performed scRNA-seq on single-cell-sorted TAME-enriched A/PB1$_{498-505}$$^+$CD8$^+$ T-cell populations from two donors with oversized populations (non-LIFT 8 and 12) and two donors with conventional-size populations (non-LIFT 14 and 10) (Fig. 7d). Unsupervised hierarchical clustering analysis revealed three gene clusters (Fig. 7e). Highly expressed genes from Cluster 1 were associated with A/PB1$_{498}$$^+$CD8$^+$ T$_{Naive}$ cells predominantly from donor 14. Most notably, A/PB1$_{498}$$^+$CD8$^+$ T$_{EMRA}$ cells from donors 8 and 12 (oversized population) were grouped together and highly expressed genes from Cluster 2, characterized by high levels of T-cell effector genes NKG7, GNLY, CCL5, and granzymes B and H (GZMB and GZMH) but not K (GZMK), found in Cluster 3. In contrast, A/PB1$_{498-505}$$^+$CD8$^+$ T$_{EMRA}$ cells from donors 14 and 8 were grouped with highly expressed genes from Cluster 1 and 3, but not Cluster 2 except for CCL5 and IFITM1 genes, revealing

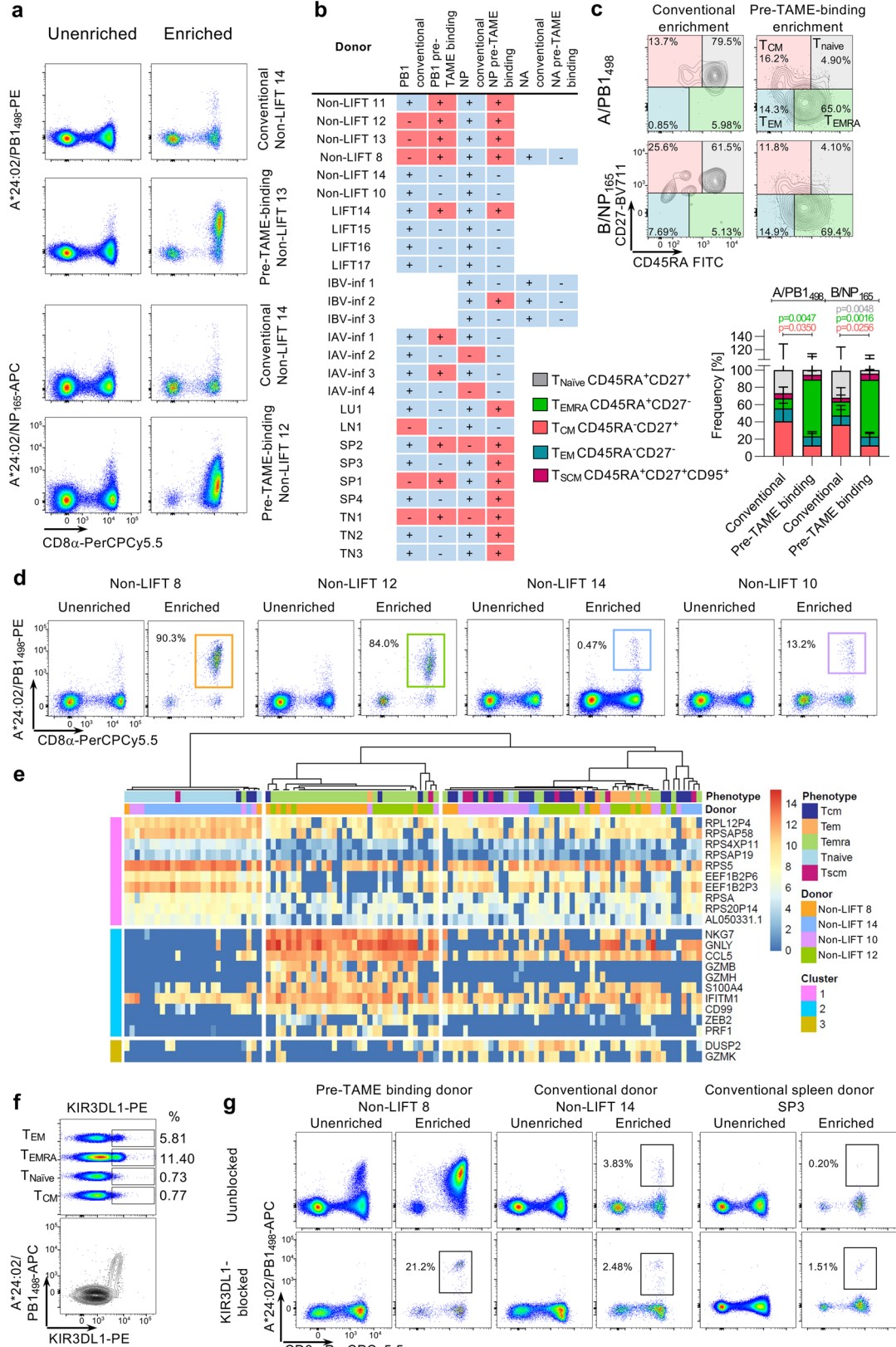

distinct characteristics of A/PB1$_{498}$+CD8+ T$_{EMRA}$ cells within the two tetramer-binding populations.

NKG7 (natural killer cell granule protein 7) and GNLY (granulysin) are key CD8+ T-cell effector genes[53] located on the same immunoregion locus containing all the natural killer-receptor genes including the killer cell immunoglobulin-like receptors (KIR), within the leukocyte receptor complex (1 Mb, chromosome 19q13.4)[54]. Since NKG7 and GNLY were the top-hit genes associated with A/PB1$_{498}$+CD8+ T$_{EMRA}$ cells from donors 8 and 12, we hypothesized that a KIR was interacting with the peptide/

**Fig. 7 Comparison of conventional and KIR-binding CD8$^+$ T-cells. a** Staining pattern of donors with conventional-size tetramer staining and donors with oversized tetramer$^+$CD8$^+$ T cell populations for the A/PB1$_{498-505}$ and B/NP$_{165-173}$ tetramers pre- and post-enrichment. **b** Presence (+) or absence (−) of conventional (blue) and oversized tetramer$^+$ T cell populations (red) across all samples tested. **c** Phenotypic analysis of tetramer-enriched CD8$^+$ T-cells from conventional-size and oversized populations for both A/PB1$_{498-505}$ and B/NP$_{165-173}$ tetramers (conventional A/PB1$_{498}$-binding cells $n = 7$ biologically independent samples; pre-TAME A/PB1$_{498}$-binding cells $n = 6$ biologically independent samples, conventional B/NP$_{165}$-binding cells $n = 9$ biologically independent samples, pre-Tame B/NP$_{165}$-binding cells $n = 7$ biologically independent samples). Bars indicate mean + SD. Statistical significance was calculated using a two-tailed Mann–Whitney test. **d** Tetramer-staining pattern of conventional and oversized-binding staining for scRNA-seq pre- and post-enrichment. **e** Unsupervised clustering of mRNA expression of enriched tetramer-binding CD8$^+$ T-cells ($n = 30$ cells per donor). All listed genes in the SC3 plot have a $p$-value < 0.05 and AUC > 0.65. $p$-values have been calculated using the two-tailed Wilcoxon signed-rank test without adjustments. **f** KIR3DL1 expression on different CD8$^+$ T-cell phenotypes in non-LIFT 8 (top panel) and co-staining of A/PB1$_{498-505}$ tetramer and KIR3DL1 without enrichment. **g** Effects of KIR3DL1 blocking on conventional and pre-TAME-binding tetramer$^+$CD8$^+$ T-cells pre- and post-enrichment.

HLA-A24 complex. KIR are expressed by a proportion of CD8$^+$ T-cells[55] and KIR3DL1, in particular, has been previously shown to bind some but not all A24 pMHC tetramers[56] implying a degree of selectivity in the interaction. Staining for KIR3DL1 revealed its expression on CD27$^-$CD8$^+$ T-cells, with the highest frequency of KIR3DL1$^+$ cells detected in the T$_{EMRA}$ population in a donor that exhibited strong Pre-TAME tetramer-binding (Fig. 7f). Co-staining with the A/PB1$_{498-505}$ tetramer showed that all tetramer-binding CD8$^+$ T-cells were positive for KIR3DL1, indicating that KIR3DL1 could potentially be binding to the tetramers (Fig. 7f). Blocking of KIR3DL1 prior to tetramer-staining markedly reduced the oversized population after TAME enrichment, to the levels of conventional tetramer$^+$CD8$^+$ T cell pools, revealing the true A24/PB1$_{498}$-specific CD8$^+$ T-cell population (Fig. 7g). Thus, much of the oversized population comprises tetramer-binding KIR3DL1$^+$CD8$^+$ T-cells with other TCR specificities. Future studies are needed to understand whether KIR3DL1 binding of peptide-HLA-A24 complexes are competing with TCR interactions to mount robust peptide-HLA-A24-specific CD8$^+$ T-cell responses, thus impacting influenza-specific immunity in Indigenous and non-Indigenous HLA-A24-expressing people at risk of severe influenza disease.

## Discussion

Indigenous populations worldwide are highly affected by pandemic and, to a lesser degree, seasonal influenza disease. In line with previous studies[57–60], our results show a high frequency of HLA-A24 allele expression in Indigenous populations in the Pacific region, an allele identified as an influenza mortality-associated allomorph[27]. In our cohort of Indigenous Australians, 36% of individuals expressed at least one HLA-A24 allele. With little understood about the nature of the HLA-A24-restricted influenza-specific CD8$^+$ T-cell response, there was a need to identify the breadth of influenza CD8$^+$ T-cell epitopes for this at-risk population. Our analysis of previously published epitopes revealed a small number of HLA-A24-restricted IAV epitopes reported as immunogenic targets, while no IBV targets for HLA-A24 were known. Our in-depth mass-spectrometry-based immunopeptidomics approach defined the breadth of peptides presented by HLA-A*24:02 during IAV and IBV infection (Supplementary Tables 2 and 3) and provided important insights into the characteristics of the associated CD8$^+$ T-cell responses that could predispose to more severe influenza disease. It should be noted that the peptidomes identified by our approach are characteristic of C1R cells (derived from EBV-transformed B cells[33]), which constitutively express immunoproteasomes. Although the presence of reactive T cells confirms the relevance of identified peptides, there is a possibility that some immunogenic peptides may be missing from our analysis due to tissue-specific processing. Of the 52 peptides presented by HLA-A*24:02 during IAV infection, most mapped to PB2 (18 peptides, 35%) and PB1 (14 peptides, 27%) viral proteins, with no peptides

originating from NA or M1. Consistent with this preference for PB2 and PB1 peptides, the CD8$^+$ T-cell response in IAV-infected HHD-A24 mice focused mostly on four epitopes from PB1 (PB1$_{216-222}$ and PB1$_{498-505}$; 36% primary splenic response) and PB2 (PB2$_{549-557}$ and PB2$_{549-559}$; 31% primary splenic response). In HLA-A*24:02$^+$ donors, memory responses to overlapping peptides PB1$_{496/498-505}$ were consistently observed, while interesting differences were seen in the hierarchy of other epitope-specific responses, with Indigenous donors responding to PA$_{649-658}$ and NP$_{39-47}$, and non-Indigenous donors instead responding to the PB2$_{549-557}$ peptide. Such differential response characteristics, possibly related to HLA co-expression or infection history, are important considerations for the design of T-cell vaccines for high-risk Indigenous populations. Importantly, CD8$^+$ T-cells specific for the dominant A/PB1$_{498-505}$ peptide were identified with an activated phenotype in the blood of patients with acute IAV infection and across different human tissues, including a population of T$_{RM}$ cells in the lung, providing evidence of their involvement in the influenza-specific response.

Hertz et al. previously showed that HLA-A24 has a low targeting efficiency for conserved regions of the pH1N1 virus, which was indicative of low cross-reactive memory responses that may have contributed to the impaired pH1N1 CD8$^+$ T-cell immunity observed in HLA-A24$^+$ individuals during the 2009 pandemic[27]. Our data reveal that HLA-A24 presents a breadth of peptide ligands (52 peptides) derived from 6/8 IAV proteins. Notably, the variable HA and NA viral glycoproteins play a minimal role in HLA-A24-restricted CD8$^+$ T-cell immunity to IAV. Instead, the focus on epitopes from PB1 and PB2 that are well conserved across virus strains circulating in South-East Asia and Australia suggests that the prominent HLA-A24-restricted CD8$^+$ T-cell responses are likely to confer broad cross-reactive immunity to IAV. This is of key importance as the current T-cell vaccines in clinical trials focus mainly on structural proteins like NP, M1, and M2, and would therefore not elicit cross-protective CD8$^+$ T-cell responses in HLA-A24$^+$ individuals at risk of severe influenza disease.

In contrast to IAV, the protein origins of IBV peptides presented by HLA-A24 differed greatly. From 48 IBV-derived presented peptides, the majority originated from NP (10, 21% of total), as well as HA (11, 23% of total) and NA (9, 19% of total with a total of 42% for surface glycoproteins). In terms of immunogenicity, our data from transgenic mice showed that the immunogenic HLA-A24-binding peptides were predominantly derived from NP (40% of response) and NA (40% of the response). More importantly, numbers of CD8$^+$ T-cells directed towards our novel epitopes were preserved during the secondary IBV challenge, indicating optimal memory establishment and recall, which contrasted with the situation for the secondary IAV challenge. As in mice, HLA-A24-restricted influenza-specific CD8$^+$ T-cell responses in Indigenous and non-Indigenous human donors were also targeted towards NP, with NP$_{165-173}$

and NP$_{164–173}$ being prominent CD8$^+$ T-cell specificities alongside CD8$^+$ T-cell epitopes derived from NA, HA, PB2, and PA (Table 3). The breadth of the HLA-A24-restricted IBV response highlights the power of identifying epitopes with our mass-spectrometry-based immunopeptidomics approach and might explain, at least partially, why Indigenous populations have not been reported to be at risk from severe IBV disease. As for IAV, IBV epitope-specific CD8$^+$ T-cells were activated during acute IBV infection in HLA-A24$^+$ individuals and were found distributed across tissues including the lung in non-infected individuals.

Broadly cross-reactive CD8$^+$ T-cell responses that provide universal immunity across multiple strains or subtypes of influenza viruses have a crucial role in protection from severe influenza disease[24]. Here we demonstrate cross-reactive responses between IBV lineages for the B/NP$_{165–173}$ peptide, as well as cross-reactive IAV/IBV responses between the A/PB2$_{549–557}$ peptide and IBV PB2$_{550–558}$ variants in HHD-A24 transgenic mice (Supplementary Fig. 11) and humans. The A/PB2$_{549–557}$ peptide is conserved between H3N2 and H1N1 IAVs[61], and shares 55% amino acid identity with the cross-reactive IBV PB2$_{550–558}$ variants. Structures of HLA-A24 with A/PB2$_{549–557}$ and B/PB2$_{550–558}$ showed that the antigen-binding cleft and both peptides adopted a similar conformation, providing a structural basis for T-cell cross-reactivity between these epitopes. Interestingly, IBV was more effective than IAV at expanding cross-reactive CD8$^+$ T-cells, suggesting that infection history may play a role in determining patterns of cross-reactivity and that selection of peptide sequences that promote the greatest cross-reactivity is a consideration for universal influenza T-cell vaccines.

Structural analysis of the overlapping peptides A/PB2$_{549–557/559}$ and B/NP$_{164/165–173}$ showed that despite the difference of only two or one amino acids in length respectively, these peptides each adopted different conformations with HLA-A24 and are likely to induce distinct TCR repertoires. In the case of B/NP$_{164/165–173}$, responses to both peptides are equivalently immunodominant in HLA-A24$^+$ individuals, providing breadth to the overall CD8$^+$ T-cell response. However, only the A/PB2$_{549–557}$ epitope showed immunogenicity in HLA-A24$^+$ individuals, with the instability and bulged conformation of the longer A/PB2$_{549–559}$ epitope potentially proving challenging for TCR recognition. Such intricacies in epitope presentation and CD8$^+$ T-cell recognition offer opportunities to either maximize or tailor responses through vaccination.

Our present study not only provides comprehensive data on generating CD8$^+$ T-cell immunity against severe influenza disease in HLA-A24-expressing Indigenous and non-Indigenous people worldwide but also unravels three potential reasons why HLA-A24 has a role in disease susceptibility: (i) *The antigenic origin of HLA-A24 IAV epitopes.* The majority (62%) of IAV peptides presented by HLA-A24 are derived from PB1 and PB2, which is in stark contrast to previous studies in humans[52,62–64] and mice[65] showing that immunodominant peptides for other HLAs are derived predominantly from NP, PA, or M1. This is problematic for the current vaccine candidates in clinical trials which do not have a PB1 or PB2 component[48,66,67], and also raises the question of whether PB1 or PB2-specific CD8$^+$ T-cell responses are equivalently robust and protective compared to immunodominant responses derived from NP, PA or M1 presented by other HLAs. (ii) *Qualitative deficiencies in the HLA-A24 IAV-specific CD8$^+$ T-cell response.* In HHD-A24 mice, the magnitude and breadth of IAV-specific CD8$^+$ T-cell responses were greatly reduced during secondary IAV (but not IBV) challenge compared to primary infection, implicating possible functional defects at memory establishment or recall levels. (iii) *Non-epitope-specific binding of peptide/HLA-A24 complexes to KIR.* This can possibly limit TCR recognition and thus TCR-specific activation of influenza-specific CD8$^+$ T-cells and could also provide an explanation for increased susceptibility to other infectious diseases such as sepsis and tuberculosis observed in Indigenous populations[14,15]. The above observations provide a platform for further investigations to understand the mechanisms driving a greater risk of severe influenza disease in HLA-A24$^+$ individuals.

Our findings provide important insights into the design of new T-cell-targeted vaccines and immunotherapy protocols to reduce influenza disease mortality and morbidity in Indigenous people globally. Current seasonal influenza vaccines are effective at inducing antibody responses to the currently circulating strains, however, they do not protect against newly emerging pandemic viruses[68,69]. It was shown previously that the inactivated influenza vaccines do not induce CD8$^+$ T-cell immunity[70]. Given the potential that CD8$^+$ T cells have to protect against pandemic influenza viruses[18,20], vaccinations are clearly not harnessing the full power of the immune system, especially against pandemic viruses to which Indigenous populations are highly susceptible. Development of a vaccine that induces long-lasting broadly cross-reactive CD8$^+$ T-cell immunity would provide at least some level of protection against distinct influenza variants, even strains with pandemic potential. Such a vaccine would minimize influenza-related deaths in global populations, especially high-risk groups, which includes HLA-A24-expressing Indigenous and non-Indigenous people. HHD-A24 transgenic mice are a valuable tool for identifying immunogenic epitopes following virus infection in an animal system in vivo. However, as there are limitations to using mono-chain HLA-transgenic mice, our study infers the immunogenic epitopes in the context of HLA-A24 allele in mice. These CD8$^+$ T cells were, however, further screened and validated in human PBMCs across several peptides presented by all HLA molecules from a number of different HLA-A24$^+$ donors. Thus, our comprehensive analysis of peptide presentation and immunogenicity defines the candidate IBV and IAV peptides needed for a CD8$^+$ T-cell-targeting vaccine that is effective in HLA-A24$^+$ individuals. Understanding how best to augment these key responses to confer stronger protective immunity is the next step.

## Methods

**Human blood and tissue samples.** Human experimental work was conducted according to the Declaration of Helsinki Principles and according to the Australian National Health and Medical Research Council Code of Practice. All blood and tonsil donors provided written consent prior to study participation. Lung tissues, lymph node, and spleen samples were obtained from deceased organ donors after receiving written informed consent from next-of-kin. Lungs were sourced from the Alfred Hospital's Lung Tissue Biobank. Lymph node and spleen were provided by DonateLife Victoria. Buffy packs were sourced from the Australian Red Cross Lifeblood (West Melbourne, Australia). Human experimental work was approved by the University of Melbourne Human Ethics Committee (ID 1955465.2, 1443389.4, 1441452.1 and 1954302.1), the Australian Red Cross Lifeblood Ethics Committee (ID 2015#8), the Alfred Hospital Ethics Committee (ID 280/14), Monash Health Human Research Ethics Committee (HREC) (ID HREC/15/MonH/64, RMH local reference number 2016/196), HREC of Northern Territory Department of Health and Menzies School of Health Research (ID 2012-1928) and Tasmanian Health and Medical HREC (ID H0017479). Human PBMCs and cells from tissues were isolated and cryopreserved as previously described[70]. Indigenous donors (LIFT) were recruited as described before[26].

**HLA typing and analysis of human PBMCs.** NGS HLA typing for HLA class I and class II on genomic DNA isolated from granulocytes were performed by the Victorian Transplant and Immunogenetics Service (West Melbourne, VIC, Australia). Co-expression Circos plots were generated using R V.4.0 (R Core Team, Vienna, Austria), RStudio: Integrated Development for R (RStudio, Inc., Boston, USA), and the cyclize package[71].

**Cell lines, viruses, and peptides.** Class I-reduced (C1R) B-LCL express low levels of HLA-A and B, but normal levels HLA-C*04:01[33,72]. C1R.A24 cells were

generated by transfecting C1R cells with HLA-A*24:02 in the pcDNA3.1(+)-hygro vector[34]. Cells were maintained in RF10 medium (RPMI-1640 with 10% heat-inactivated FCS, 100 mM MEM non-essential amino acids, 55 mM 2-mercaptoethanol, 5 mM HEPES buffer solution, 1 mM MEM sodium pyruvate, 1 mM L-glutamine, 100 U mL$^{-1}$ penicillin, and 100 mg mL$^{-1}$ streptomycin (purchased from Gibco/ThermoFisher Scientific) with the addition of 0.3 mg mL$^{-1}$ hygromycin-B (Life Technologies) for C1R.A24 cells. Influenza A (A/HKx31 & A/Puerto Rico/8/1934 (A/PR8)) and B viruses (human, non-mouse adapted isolates; B/Malaysia/2506/04 & B/Phuket/3073/2013) were grown for 3 days at 35 °C in the allantoic cavity of 10 day-old embryonated chicken eggs. Viral titers were determined by performing semisolid overlay plaque assay on MDCK cells (ATCC). Influenza B viruses were kindly provided by Steve Rockman (Seqirus, Australia). Influenza peptides were synthesized by GenScript (Piscataway, NJ, USA) with a purity > 80% and reconstituted at 1 mM in 100% DMSO. Cell lines tested negative for mycoplasma by PCR (Supplementary Table 7).

**Expansion of antigen-specific memory CD8$^+$ T-cells from human PBMC.** Cryopreserved PBMCs (3.3–5 × 10$^6$) from healthy non-LIFT and LIFT donors were used to expand antigen-specific CD8$^+$ T-cells modified from Koutsakos et al.[24]. In brief, one-third of PBMCs were pulsed with a pool of up to 31 peptides (including circulating variants, Supplementary Table 4) at a total concentration of 10 μM at 37 °C in RPMI. After 1 h, cells were washed twice with RPMI and mixed with the remaining autologous PBMCs. To expand virus-specific CD8$^+$ T-cells, infected C1R.A24 cells were washed twice with serum-free RPMI to remove excess FCS and infected with A/HKx31 or B/Malaysia/2560/2004 at a MOI of 5 and incubated at 37 °C. After 1 h, RF10 was added and cells were incubated for further 11 h at 37 °C before cells were placed at 4 °C for 14 h. Infected C1R.A24 cells were washed twice to remove any residual virus and added at a 1:10 ratio to PBMCs. For additional stimulation, virus-expanded PBMCs were restimulated by the addition of virus-infected C1R.A24 cells on day 8 at a 1:10 ratio. Cells were then incubated for a total of 10–15 days in RF10 media with 10 U mL$^{-1}$ of recombinant human IL-2 (Roche Diagnostics, Mannheim, Germany) being added on day 4 and half-media changes every 1–2 days onwards.

**T-cell restimulation and intracellular cytokine staining.** To identify epitope-specific CD8$^+$ T-cells that expanded after stimulation, cells from days 10–15 cultures (2 × 10$^5$ cells/well) of peptide-expanded PBMCs were mixed with peptide-pulsed C1R.A24 at a 1:3 ratio while virus-expanded PBMCs were restimulated by direct peptide addition (1 μM). Cells were incubated for 5 h in the presence of Brefeldin A (BD Golgi Plug), Monensin (BD Golgi Stop) and anti-CD107a FITC at 37 °C. Cells were stained with panel 2 (Supplementary Table 6) and analyzed using flow cytometry (BD Fortessa) and FlowJo v10 (BD).

**Large-scale infection for immunopeptidome analysis.** For large-scale infections, C1R or C1R.A24 were cultured to high density in RF10 media slowly rotating in 17 dm$^2$ filter-capped roller bottles (Corning) at 37 °C, 5% CO$_2$. Cells were harvested and infected with influenza A or B virus at a MOI of 5 in RPMI at a density of 1 × 10$^7$ cells mL$^{-1}$ in 50 mL tubes for 1 h at 37 °C with slow rotation. Infected cells were returned to roller bottles with the addition of 1:1 conditioned media:RF10 to a final density of 1.4 × 10$^6$ cells mL$^{-1}$ and incubated a further 1–15 h (37 °C, 5% CO$_2$, slow rotation). HLA expression and infection efficacy were validated by surface staining ~10$^6$ cells with anti-HLA class I PE-Cy7 (1:200 in PBS; clone w6/32, Biolegend, Cat# 311430; 30 min, 4 °C), prior to washing in PBS, fixation in 1% paraformaldehyde (ProSciTech) in PBS (20 min, room temperature), and intracellular staining with anti-NP FITC for influenza A (Clone 1331, GeneTex Cat# GTX36902) or influenza B (Clone H89B, ThermoFisher Cat# MA1- 7306) (1:200 in 0.3% saponin [Sigma] in PBS, 30 min, 4 °C). Cells were washed in PBS and acquired by flow cytometry using a BD LSRII flow cytometer running BD FACSDiva software, and analyzed using FlowJo_v10 (BD). The remaining cells were harvested by centrifugation in 500 mL V-bottom flasks (3283 × g, 15 min, 4 °C), washed in PBS, snap-frozen as cell pellets in liquid nitrogen, and stored at −80 °C until use. Uninfected cells were harvested and snap-frozen as for infected cells.

**LC-MS/MS analysis of HLA-bound peptides.** Cell pellets of 0.7–1.3 × 10$^9$ C1R or C1R.A24 were lysed via cryogenic milling (Retsch Mixer Mill MM 400), resuspension in 0.5% IGEPAL CA-630, 50 mM Tris-HCl pH 8.0, 150 mM NaCl and protease inhibitors (cOmplete Protease Inhibitor Cocktail Tablet; Roche Molecular Biochemicals) and incubation at 4 °C for 1 h with slow rotation. Lysates were cleared by ultracentrifugation and HLA isolated by immunoaffinity purification using protein-A-agarose-bound antibodies as described[35,73]. Antibodies were either w6/32 (pan class I) alone or sequential DT9 (HLA-C specific), w6/32 (pan class I), and mixed class I (equal amounts LB3.1, SPV-L3 and B721, capturing HLA-DR, -DQ and -DP, respectively).

Peptide/MHC complexes were dissociated and fractionated by reversed-phase high-performance liquid chromatography (RP-HPLC) as described[24,35,74]. 500 μL fractions were collected throughout the gradient, and the peptide containing fractions combined into 9 pools, vacuum-concentrated, and reconstituted in 15 μL 0.1% formic acid (Honeywell) in Optima™ LC-MS water. Reconstituted fraction

pools were analyzed by LC-MS/MS using a SCIEX 5600+ TripleTOF mass spectrometer equipped with a Nanospray III ion source as previously described[74].

**LC-MS/MS data analysis.** Spectra were searched against a proteome database consisting of the human proteome (UniProt/Swiss-Prot v2016_04), and either the A/X31 or the B/Malaysia/2506/2004 proteome plus a 6-reading frame translation of the viral genome, using ProteinPilot software (version 5.0, SCIEX), considering biological modifications and employing a decoy database to calculate the false discovery rate (FDR). Subsequent analyses were based on the best hypothesis for distinct peptides. Sequence motifs were generated utilizing peptides assigned at confidences greater than that required for a 5% FDR using Seq2logo2.0[75] (default settings). Likely HLA-A*24:02 binders were determined based on appearance across the experiments/antibodies and predicted binding (netMHCpan4.0[39–41]). For peptides identified in their native form (and lacking Cys residues) that were synthesized for functional analysis, fragmentation patterns and retention times of representative spectra were compared to the synthetic and the quality of the match described (Supplementary Data 1).

**HLA-A*24:02 HHD mouse studies.** All mouse studies were overseen by the University of Melbourne Ethics Committee (#171408). HHD-A24 mice were generated by François Lemonnier as described previously[42]. These mice express a chimeric MHC-I that consists of the murine α3 and transmembrane domain and the human α1 and α2 domain covalently linked the human β2m. For mouse infections, 6–12 week-old mice were infected intranasally using a positive displacement pipette with 30 μL of either 100 pfu of A/X-31 or 200 pfu of B/Malaysia/2506/2004 in PBS under isoflurane anesthesia. For the secondary challenge, mice were infected 6–8 weeks after primary infection with 200 pfu of A/PR8 or 400 pfu B/Phuket/3073/2013. To identify immunogenic peptides, spleen and bronchioalveolar lavage (BAL) were isolated on day 10 or day 8 for secondary infection, respectively. Spleen single-cell suspensions were prepared and incubated for 1 h at 37 °C in Affinipure Goat anti-mouse IgG + IgM (Jackson Immunoresearch)-coated panning plates to deplete B cell populations. BAL was combined from 3 to 5 mice to achieve sufficient T-cell numbers. Single-cell suspensions were then stimulated with peptide pools or single peptides at 1 μM in the presence of Brefeldin A (BD Golgi plug) for 5 h at 37 °C in RF10 with 10 U mL$^{-1}$ IL-2 followed by staining with panel 1 (Supplementary Table 6). Cells were analyzed using BD Fortessa and FlowJo v10 (BD). For immunization studies, mice were vaccinated with 30 nmol of NA$_{32–40}$, NP$_{392–400}$ and NP$_{165–173}$ emulsified in complete (Prime) or incomplete (Boost) Freund's adjuvant. 50 μL vaccine was injected on both sides at the base of the tail. Control mice were injected emulsified adjuvant without peptides. Two weeks after prime, mice were boosted and challenged with 1 × 10$^3$ pfu B/Malaysia/2506/2004 intranasally 7 days after boost. On days 6 and 7, after infection lungs were isolated to determine viral load with a plaque assay as described before[76].

**Detection of cytokines in BAL using a cytometric bead array.** Cytokine concentrations in the BAL were detected using the BD Cytometric Bead Array Mouse enhanced sensitivity master buffer kit (Cat#: 562248) as per the manufacturer's instruction. Data were acquired using BD FACS Canto II and analyzed by FCAP Array software (Soft Flow Inc. Pecs, Hungary).

**Protein expression, purification, and crystallization.** Soluble HLA-A*24:02 heterodimers containing either A/PB2$_{549–557}$, A/PB2$_{549–559}$, B/PB2$_{549–557}$, B/NP$_{165–173}$, or B/NP$_{164–173}$ peptide were prepared as follows. A truncated HLA-A*24:02 construct encompassing the extracellular part of the HLA molecule (residues 1–276), and human beta-microglobulin (β2 m) were expressed separately in a BL21-pLyS *Escherichia coli* strain as inclusion bodies. The inclusion bodies were subsequently extracted, washed, and resuspended into a solution containing 6 M guanidine. Each pHLA complex was then refolded into a cold refolding solution (100 mM Tris-HCl pH 8, 2 mM Na-EDTA, 400 mM L-arginine-HCl, 0.5 mM oxidized glutathione, 5 mM reduced glutathione) by adding 30 mg of HLA heavy chain, 20 mg of β2 m and 4 mg of peptide. The refolding solution was then dialyzed in 10 mM Tris-HCl pH 8, and the protein was purified by a succession of affinity column chromatography.

**Crystallization, data collection, and structure determination.** Crystals of the pHLA-A*24:02 complexes were grown by the hanging-drop, vapor-diffusion method at 20 °C with a protein/reservoir drop ratio of 1:1 with seeding at a concentration of 6 mg mL$^{-1}$ in the following with conditions: A/PB2$_{549–557}$: 20% PEG3350, 0.2 M Na acetate; PB2$_{549–559}$: 24% PEG8K, 0.1 HEPES pH 7.5, 5% v/v Ethyl acetate; B/PB2$_{549–557}$: 24% PEG8K, 0.1 HEPES pH 7.5, 2% isopropanol, 5% w/v PolyvinylpyrrolidoneK15; B/NP$_{165–173}$: 19% 3350, 0.2 MgCl$_2$; NP$_{164–173}$: 20% PEG8K, 0.2 MgCl$_2$, 0.1 Tris-HCl pH 8.5. The crystals were soaked in a cryoprotectant solution containing mother liquor solution with the PEG concentration increased to 30% (w/v) and then flash-frozen in liquid nitrogen. The data were collected on the MX1 and MX2 beamlines[77]. Manual model building was conducted using the Coot software[78] followed by maximum-likelihood refinement with the Buster program[79]. The final models were validated using the Protein Data Bank validation website and the final refinement statistics are summarized in

Supplementary Table 5. The final models were deposited in the PDB with the accession codes: 7JYU (HLA-A*A2402-NP 164-173); 6XQA (HLA-A*24:02-PB2549-557B); 7JYV (HLA-A*A2402-NP 165-173); 7JYW (HLA-A*24:02-PB2549-557); 7JYX (HLA-A*24:02-PB2549-559). All molecular graphics representations were created using PyMol[80].

**Thermal stability assay**. Thermal shift assays were performed to determine the stability of each pHLA-A*24:02 complex using fluorescent dye Sypro orange to monitor protein unfolding. The thermal stability assay was performed in the Real Time Detection system (Corbett RotorGene 3000), originally designed for PCR. Each pHLA complex was in 10 mM Tris-HCl pH 8, 150 mM NaCl, at two concentrations (5 and 10 mM) in duplicate, was heated from 25 to 95 °C with a heating rate of 1 °C min$^{-1}$. The fluorescence intensity was measured with excitation at 530 nm and emission at 555 nm. The Tm, or thermal melt point, represents the temperature for which 50% of the protein is unfolded.

**Tetramer-associated magnetic enrichment in humans**. TAME was performed on PBMCs ($7.5 \times 10^6$–$2.7 \times 10^8$) of healthy, IAV- or IBV-infected donors, as well as lymphocytes isolated from tonsils, lung, and pancreatic lymph nodes (panLN) to detect CD8$^+$ T-cells specific for IAV and IBV. pMHC-I monomers were made in-house[81] and conjugated at an 8:1 molar ratio to PE- or APC-labeled streptavidin (SA) to generate tetramers. Cells were FcR-blocked for 15 min on ice and stained with APC or PE-conjugated tetramers at a 1:100 dilution for 1 h at RT, washed twice then incubated with anti-PE and anti-APC MicroBeads (Milenty Biotec). Unenriched, flow-through, and enriched fractions were surface stained with panel 3 (PBMC) or 4 (SLO and lung) (Supplementary Table 6). After 30 min staining on ice, cells were fixed for 20 min in 1% PFA and acquired by flow cytometry. In some experiments, KIR3DL1 blocking was achieved by the addition of anti-human NKB1 antibody (clone DX9, Cat 555964, BD Pharmingen) at 1:50 during the FcR-blocking step.

**Single-cell mRNAseq**. A/PB1$_{498-505}$$^+$ CD8$^+$ T-cells were single-cell sorted into 96 well plates containing lysis buffer (1 μL RNase inhibitor and 19 μL Triton X-100) after TAME on a BD Aria III sorter. Libraries were generated as described previously[24]. A Nextera XT DNA Library Prep Kit was used for the generation of sequencing libraries and sequencing performed on a NextSeq500 platform with 150-base par high-output paired-end chemistry for 30 tetramer$^+$ cells/donor (120 cells total).

**Bioinformatical analysis**. The quality of scRNA-seq was assessed with FastQC. TopHat2 with default parameters was used to align sequences to the Ensembl GRCh38 reference genome. A total of 112 out of 120 analyzed cells passed quality control and was used for further analysis. Gene expression was quantified utilizing Cufflinks suit (v 2.2.1) where FPKM was assessed using CuffQuant and values normalized based on total mRNA content with CuffNorm. Clustering was performed utilizing SC3[82]. Each gene has been assessed for the accuracy of predicting the 3 clusters using its expression level. The accuracy of the prediction has been measured using the area under the receiver operating characteristic (ROC). Furthermore, a p-value has been calculated using the two-tailed Wilcoxon signed-rank test. Pheatmap in R was used to visualize Heatmaps.

**Statistical analyses**. Statistical analysis was performed using Graphpad Prism (v8.4.2, Graphpad, USA). Non-parametric unpaired data was analyzed performing Mann–Whitney test whereas paired data was analyzed with Wilcoxon matched-pairs signed-rank test. Two-tailed analysis was performed with $p < 0.05$ (*) or $p < 0.01$ (**) indicated.

**Reporting summary**. Further information on experimental design is available in the Nature Research Reporting Summary linked to this paper.

## Data availability

Mass spectrometry proteomics data have been deposited in the ProteomeXchange Consortium via the PRIDE[83] partner repository under the accession code PXD020292. scRNA-seq data that support the study are deposited in Arrayexpress with accession E-MTAB-10109. All crystal structures are deposited 7JYU (HLA-A*A2402-NP 164-173); 6XQA (HLA-A*24:02-PB2549-557B); 7JYV (HLA-A*24:02-PB2549-557); 7JYX (HLA-A*24:02-PB2549-559). Influenza virus sequences are available on the influenza virus database [https://www.ncbi.nlm.nih.gov/genomes/FLU/Database/nph-select.cgi?go=database]. All other data that support the findings of this study are present in the article and its Supplementary Information files or from the corresponding author upon request. Source data are provided with this paper.

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

## Acknowledgements
HHD-A24 transgenic HHD mice were developed by Dr. François Lemonnier (Pasteur Institute, Paris, France). We thank the Monash Macromolecular Crystallization Facility staff and the staff at the Australian synchrotron for technical assistance. This research was undertaken in part using the MX2 beamline at the Australian Synchrotron, part of ANSTO, and made use of the Australian Cancer Research Foundation (ACRF) detector. The Australian National Health and Medical Research Council (NHMRC) Program Grant (#1071916) to K.K., NHMRC Project Grant (#1122524) to K.K., S.T., A.M., S.G.,

and A.W.P., and NHMRC Investigator Grant (#1173871) to K.K. supported this work. This work was also supported by NIAID UO1 grant 1U01AI144616-01; Dissection of Influenza Vaccination and Infection for Childhood Immunity (DIVINCI) to K.K. and by the ARC grant # DP190103282 to K.K., A.G.B., and L.L. L.H. was a recipient of Melbourne International Research Scholarship and Melbourne International Fee Remission Scholarship. C.E.S. had received funding from the European Union's Horizon 2020 research and innovation program under the Marie Skłodowska-Curie grant agreement (#792532). J.R. is supported by an ARC Laureate fellowship. S.G. is an NHMRC SRF-A Fellow (#1159272). E.B.C. is NHMRC Peter Doherty Fellow. A.W.P. is supported by an NHMRC Principal Research Fellowship (#1137739) and NHMRC Project grant (#1085018) to A.W.P., N.A.M., and T.C.K. S.T. is an NHMRC Career Development Fellow (#1145033). E.J.G. is an NHMRC CJ Martin Fellow. P.T.I. was supported by an NHMRC Early Career Fellowship (#1072159) and Monash University Faculty of Medicine, Nursing and Health Sciences Senior Postdoctoral Fellowship (2020).

## Author contributions

L.H., P.T.I., E.B.C., T.H.O.N., C.E.S., L.L., N.A.M., S.G., J.R., and K.K. designed experiments. L.H., P.T.I., E.B.C., T.H.O.N., M.K., C.E.S., N.A.M., A.T.N., C.S., H.H., E.J.G., L.L., B.C., and S.G. performed experiments. S.R., T.C.K., A.C.C., M.R., G.P.W., L.M.W., T.L., S.I.M., M.E., S.G.T., J.D., A.M., S.Y.C.T., P.S., K.F., D.C.J., and A.B. provided reagents and/or samples. L.H., P.T.I., E.B.V., T.H.O.N., N.A.M., A.T.N., S.R., F.L., J.R., S.G., A.W.P., and K.K. analyzed data. L.H., T.H.O.N., P.T.I., S.G., and K.K. wrote the manuscript. All authors read and approved the manuscript.

## Competing interests

S.R. is an employee of Seqirus Ltd. L.H., E.B.C., M.K., and K.K. are named as co-inventors in a patent application filed by the University of Melbourne (PCT/AU2018/050971) covering the use of some peptides described in the publication as part of vaccine formulation. The remaining authors declare no competing interests.

## Additional information

[1]Department of Microbiology and Immunology, University of Melbourne, at the Peter Doherty Institute for Infection and Immunity, Parkville, VIC, Australia. [2]Department of Biochemistry and Molecular Biology & Infection and Immunity Program, Biomedicine Discovery Institute, Monash University, Clayton, VIC, Australia. [3]Department of Hematopoiesis, Sanquin Research and Landsteiner Laboratory, Amsterdam UMC, University of Amsterdam, Amsterdam, The Netherlands. [4]School of Medical Sciences and The Kirby Institute, UNSW Sydney, Sydney, NSW, Australia. [5]Seqirus, Parkville, VIC, Australia. [6]Department of Allergy, Immunology and Respiratory Medicine, The Alfred Hospital, Melbourne, VIC, Australia. [7]Department of Medicine, Monash University, Central Clinical School, The Alfred Hospital, Melbourne, VIC, Australia. [8]School of Public Health and Preventive Medicine, Monash University, Melbourne, VIC, Australia. [9]Infection Prevention and Healthcare Epidemiology Unit, Alfred Health, Melbourne, VIC, Australia. [10]Victorian Infectious Diseases Service, The Royal Melbourne Hospital, and Doherty Department University of Melbourne, at the Peter Doherty Institute for Infection and Immunity, Parkville, VIC, Australia. [11]Lung Transplant Unit, Alfred Hospital, Melbourne, VIC, Australia. [12]Immunology and Diabetes Unit, St Vincent's Institute of Medical Research, Fitzroy, VIC, Australia. [13]Sydney Medical School, University of Sydney, Sydney, NSW, Australia. [14]Chris O'Brien Lifehouse Cancer Centre, Royal Prince Alfred Hospital, Sydney, NSW, Australia. [15]Immunology Division, Garvan Institute of Medical Research, Darlinghurst, NSW, Australia. [16]St. Vincent's Clinical School, University of New South Wales, Sydney, NSW, Australia. [17]Tasmanian Vaccine Trial Centre, Launceston General Hospital, Launceston, TAS, Australia. [18]School of Health Sciences and School of Medicine, University of Tasmania, Launceston, TAS, Australia. [19]Department of Immunology and Pathology, Monash University, Melbourne, VIC, Australia. [20]School of Health and Biomedical Science, RMIT University, Melbourne, VIC, Australia. [21]Institute of Infection and Immunity, Cardiff University School of Medicine, Cardiff, UK. [22]Australian Research Council Centre of Excellence for Advanced Molecular Imaging, Monash University, Clayton, VIC, Australia. [23]Department of Biochemistry and Genetics, La Trobe Institute for Molecular Science, La Trobe University, Bundoora, VIC, Australia. [24]Menzies School of Health Research, Darwin, NT, Australia. [25]Indigenous Engagement, CQUniversity, Townsville, Australia. [26]These authors contributed equally: Luca Hensen, Patricia T. Illing, E. Bridie Clemens. [27]These authors jointly supervised this work: Steven Y.C. Tong, Anthony W. Purcell, Katherine Kedzierska. ✉email: kkedz@unimelb.edu.au

