## [Peer Review File · Nature Communications]

REVIEWER COMMENTS

Reviewer #1 (Remarks to the Author):

Previous reports have found that indigenous populations, which are enriched for the HLA A*2402 allele are at a high risk of developing severe influenza infection, but the mechanisms remain unclear. Hensen et al. present an in-depth analysis of the CD8+ T-cell landscape of A*2402 humans and transgenic mice to influenza A and B infections. This paper addresses the important question of the role of CD8+ T-cell responses in influenza infection, especially in pandemic settings where antibodies to previous influenza strains fail to cross react with the pandemic strain, highlighting the role of T-cells. While there is preliminary evidence for the protective role of CD8+ T-cells in H7N9 infections, such responses have not been previously characterized in indigenous populations. Studying such responses may be important in boosting protection in this high-risk group.

To study T-cell responses in indigenous subjects, the authors recruited a cohort of 127 participants 36% of which expressed HLA-A24. The authors first tested a few previously published A*24 T-cell epitopes but found that only 3/12 of these were positive in a subset of 5 donors from their cohort. They then used a peptidomic approach to identify peptides bound to HLA A*2402 cell lines following infection with 2 influenza strains identifying a total of 52 x31 and 48 B/Malaysia potential epitopes, which spanned the entire viral proteome.

They then used HLA A-24 transgenic mice to study in-vivo responses to these peptides following IAV infection using ICS, identifying 2 PB1 and 3 PB2 dominant responses as well as two additional subdominant responses. Most responses post 2nd challenge (PR8) were to these 6 dominant epitopes, and two responses were lost. The authors duly note that current T-cell vaccines mostly focus on responses to M1 and NP, and not PB1 and PB2 that were identified here.

The same experimental setup was also used to identify immunodominant peptides for IBV infection, leading to the identification of 3 dominant responses in NP and NA and an additional 4 sub-dominant responses, most of which were lost after 2nd infection.

They then tested responses in the human cohort, also using mutant peptides of their immunodominant epitopes generated using an online tool. Peptides were first tested in pools of 7-13 peptides each. In 5/5 donors tested responses were dominated to the PB1498-505 epitope, which was also immunodominant in mice, as well as to its longer version (PB1496-505), and also to PA649-658 and NP39-47. Two of these responses were also immunodominant in non-indigenous subjects tested, suggesting differential immunodominance hierarchies in these two different populations. They then tested pools of 41 IBV peptides identified by mass spec. and found response to a pool containing the 6 immunodominant epitopes in both indigenous and non-indigenous donors. In total 9/41 peptides were immunogenic with 3 found in both mice and humans, 3 only in humans and 3 only in mice.

To test the cross-reactive potential of the IBV responses to the Yamagata lineage, the authors restimulated with variants from this lineage. This analysis identified cross-reactive responses to two epitopes. They further identified a peptide with 55% sequence identity between IAV and IBV (IAV PB2549-557 TYQWIIRNW and IBV PB2550-558). They found cross-reactive responses to this peptide in 2/4 and 4/4 donors (tested both ways). They then solved the structure of these peptides bound to HLA A*2402 and showed that they had highly similar structures (average RMSD of 0.31Å). The structures also showed that two variants of the same peptide NP164-173 and NP165-173 were bound in different conformations.

To determine the protective effect of these CD8 T-cell responses mice were vaccinated with the 3 immunogenic IBV peptides and then challenged with IBV Malaysia. Vaccinated mice had decreased disease severity, reduced viral titers in the lungs at day 7 as well as decreases in inflammatory cytokine levels.

Tetramers were generated from the immunodominant epitopes from IAV and IBV to directly test ex-vivo in healthy and influenza infected subjects and found frequencies in line with previously

published data in IAV memory. However post-infection responses were different than those of uninfected subjects demonstrating the existence of highly activated CD8+ T-cells against the previously published A/PB1498 epitope and the IBV epitopes identified in this study.

Overall this is a great paper, very well written and easy to follow. It presents compelling evidence for the important role of CD8 T-cell responses in IAV and IBV infections, focusing on indigenous populations, which have been previously shown to be prone to severe disease during influenza pandemics. The paper has several key findings:

1. New immunodominant epitopes of IBV and IAV for A*24
2. New IBV/IAV cross reactive T-cell response including an insightful structural analysis
3. Demonstration that overlapping epitopes may bind in different conformations to the same HLA
4. Preliminary evidence that CD8 T-cell responses may be protective in a mouse model.

I therefore recommend the paper be accepted for publication in its current form. I want to commend the authors on beautifully created figures that are easy to follow. I typically have lots of quirks with figures in papers I review and these were just done very well.

Reviewer #2 (Remarks to the Author):

In the manuscript by Hensen et al. the authors define the influenza epitopes recognized by an Indigenous population in Australia that enriched in the HLA-A 24:02 allele that worldwide has a 10% global distribution. Already characterized epitopes in the general population as well as the newly described epitopes are tested for their ability to evoke functional Th1 responses from PBMC and the frequency in Indigenous and non-Indigenous populations is described. The manuscript is well written and thorough in its experimental approach. The authors argue that differences in immunodominant epitopes contribute to the increased susceptibility of the Indigenous population to severe influenza disease.

Comments on the manuscript are as follows:

1. The manuscript may be more digestible if divided into two separate manuscripts. One defining the immunodominant epitopes in the two populations and their characteristics and another following up discussing the functional implications.
2. Indigenous individuals display increased susceptibility to numerous other pathogens and APC have been implicated. Inclusion of a discussion of such information would add to the manuscript. As would inclusion of literature regarding the impact of previous infections on CD8 responses and disease outcomes.
3. The LIFT cohort information in Figure 1b focuses on females. Both male and female information should be described, particularly for HLA-A24+ proportions in both sexes.
4. All representative FACS plots with gates should include frequencies and the shading on plots in figures (Fig 6 and 7) should be removed or lightened as the contours are difficult to see.
5. Clarity is required regarding the ungated seemingly tetramer positive cells in Figure 6a NP tetramer in the spleen and tonsil. Do these cells respond to peptide stimulation and become dual positive tetramer and cytokine producing cells? If these are real cells, the data would conflict with the main hypothesis of the manuscript.
6. The authors use numerous strains of influenza in mice and should provide information regarding their mouse adapted nature, especially for those that are not as common and well-described such as X31 and PR8.
7. In the protection experiments in Figure 5, the differences in weight loss are marginal and error bars overlapping. The authors could consider using whole protein versus CD8 peptide priming as

the lack of helper epitopes will lead to the generation of helpless CD8 T cells and the findings of their formation and function as memory questionable.

8. Given that most humans are not mono chain for HLA alleles, whether or not one can make assumptions regarding dominant immunogenic epitopes using mono-chain HLA-transgenic mice is not clear. While these mice are valuable to allow one to say a response can be generated, it is questionable whether inferences about immunodominance and vaccine formulations in the absence of other competing alleles can be made.

Reviewer #3 (Remarks to the Author):

This is a well performed study, that identifies previously unknown epitopes of influenza B that are immunodominant in indigenous populations expressing the HLA A*24:02 allomorph. This result is not unexpected, since sufficient strength of binding of peptides to HLA alleles is essential for forming a T cell epitope. It is however significant because it reveals that focusing on just a few viral proteins may be a poor strategy for successfully immunizing diverse racial groups, when they express distinct HLA alleles. The study re-enforces the fact that in most individuals only a few peptides are immunodominant, also well-known from previous studies, but since this may lead to lack of vaccine-induced protection in this case, it could have more stark consequences. Thus, this is an interesting study which indicates the impact of HLA variants in indigenous people on response to different influenza epitopes.

Detailed comments:

1. This is a thorough study with nice graphics. But it is very dense and it would be useful if the authors could give more summary information as they proceed.
2. Is there more diversity in responses of non-indigenous populations or is the degree of restriction to immunodominant epitopes equivalent?
3. What evidence is there that vaccines currently in use fail to induce protection?
4. There are a number of minor spelling errors.

Reviewer #4 (Remarks to the Author):

Hensen et al. characterize the HLA-24 associated influenza A and B virus immunopeptidome and associated CD8+ T cell response in humans and HLA-24 transgenic mice to better understand the high susceptibility of indigenous expressing individuals associated with HLA-24 expression. This is an outstanding "soups to nuts" study that exploits world class expertise of the collaborating investigators in mass spectrometry, structural biology, and human mouse immunology. The addition of so many peptides to the influenza peptidome, is in itself, a major contribution to the viral immunology field.

A few comments for the authors to consider.

1. Given the presence of at least 2, and up to 3 other HLA allomorphs in HLA-24 individuals, it seems more likely that any HLA-24 role in disease susceptibility is due not to a lack of function of HLA-24 in presenting peptides (as originally proposed by Hertz et al., 2013) but rather some deleterious gain of function effect. The identification of some many peptide ligands and T cells supports this interpretation. Perhaps the authors could raise this issue and briefly propose some alternative hypotheses.
2. It would be useful for non-specialist readers to provide a brief summary of CD8+ T cell

immunodominance and a few references to relevant reviews.

3. No comments on the kinetics of peptide generation or the identification of out of reading frame peptides? Is there a relationship between peptide abundance and time of detection? The early detection of most peptides from biochemically stable proteins (all of the flu proteins except PB1-F2) strongly points to DRiPs being the major source of peptides, particularly since the frame shifted peptides appear in the same time frame as most other peptides. A few sentences at least would seem to be called for unless the authors plan expand this analysis in another paper.

4. At least one of the peptide identified, NP39 is known to be ligand for a different MHC I molecule (Kd). This ought to be mentioned and referenced, as well as any other peptides that are known to be presented by other allomorphs, as can be revealed simply by searching peptidome DBs.

5. It should be mentioned that C1R cells are B cells that constitutive express immunoproteasomes and as such, likely have a highly overlapping but distinct peptidome compared to cells expressing standard/mixed proteasomes.

REVIEWER COMMENTS

We thank immensely the Reviewers for their comments and for acknowledging the novel aspects of our study.

Reviewer #1 (Remarks to the Author):

Previous reports have found that indigenous populations, which are enriched for the HLA A*2402 allele are at a high risk of developing severe influenza infection, but the mechanisms remain unclear. Hensen et al. present an in-depth analysis of the CD8+ T-cell landscape of A*2402 humans and transgenic mice to influenza A and B infections. This paper addresses the important question of the role of CD8+ T-cell responses in influenza infection, especially in pandemic settings where antibodies to previous influenza strains fail to cross react with the pandemic strain, highlighting the role of T-cells. While there is preliminary evidence for the protective role of CD8+ T-cells in H7N9 infections, such responses have not been previously characterized in indigenous populations. Studying such responses may be important in boosting protection in this high-risk group.

To study T-cell responses in indigenous subjects, the authors recruited a cohort of 127 participants 36% of which expressed HLA-A24. The authors first tested a few previously published A*24 T-cell epitopes but found that only 3/12 of these were positive in a subset of 5 donors from their cohort. They then used a peptidomic approach to identify peptides bound to HLA A*2402 cell lines following infection with 2 influenza strains identifying a total of 52 x31 and 48 B/Malaysia potential epitopes, which spanned the entire viral proteome.

They then used HLA A-24 transgenic mice to study in-vivo responses to these peptides following IAV infection using ICS, identifying 2 PB1 and 3 PB2 dominant responses as well as two additional subdominant responses. Most responses post 2nd challenge (PR8) were to these 6 dominant epitopes, and two responses were lost. The authors duly note that current T-cell vaccines mostly focus on responses to M1 and NP, and not PB1 and PB2 that were identified here.

The same experimental setup was also used to identify immunodominant peptides for IBV infection, leading to the identification of 3 dominant responses in NP and NA and an additional 4 sub-dominant responses, most of which were lost after 2nd infection.

They then tested responses in the human cohort, also using mutant peptides of their immunodominant epitopes generated using an online tool. Peptides were first tested in pools of 7-13 peptides each. In 5/5 donors tested responses were dominated to the PB1498-505 epitope, which was also immunodominant in mice, as well as to its longer version (PB1496-505), and also to PA649-658 and NP39-47. Two of these responses were also immunodominant in non-indigenous subjects tested, suggesting differential immunodominance hierarchies in these two different populations. They then tested pools of 41 IBV peptides identified by mass spec. and found response to a pool containing the 6 immunodominant epitopes in both indigenous and non-indigenous donors. In total 9/41 peptides were immunogenic with 3 found in both mice and humans, 3 only in humans and 3 only in mice.

To test the cross-reactive potential of the IBV responses to the Yamagata lineage, the authors restimulated with variants from this lineage. This analysis identified cross-reactive responses to two epitopes. They further identified a peptide with 55% sequence identity between IAV and IBV (IAV PB2549-557 TYQWIIRNW and IBV PB2550-558). The found cross-reactive responses to this peptide in 2/4 and 4/4 donors (tested both ways). They then solved the structure of these peptides bound to HLA A*2402 and showed that they had highly similar structures (average RMSD of 0.31Å). The structures also showed that two variants of the same peptide NP164-173 and NP165-173 were bound in different conformations.

To determine the protective effect of these CD8 T-cell responses mice were vaccinated with the 3 immunogenic IBV peptides and then challenged with IBV Malaysia. Vaccinated mice had decreased disease severity, reduced viral titers in the lungs at day 7 as well as decreases in inflammatory cytokine levels.

Tetramers were generated from the immunodominant epitopes from IAV and IBV to directly test ex-vivo in healthy and influenza infected subjects and found frequencies in line with previously published data in IAV memory. However post-infection responses were different than those of un-infected subjects demonstrating the existence of highly activated CD8+ T-cells against the previously published A/PB1498 epitope and the IBV epitopes identified in this study.

Overall this is a great paper, very well written and easy to follow. It presents compelling evidence for the important role of CD8 T-cell responses in IAV and IBV infections, focusing on indigenous populations, which have been previously shown to be prone to severe disease during influenza pandemics. The paper has several key findings:

1. New immunodominant epitopes of IBV and IAV for A*24
2. New IBV/IAV cross reactive T-cell response including an insightful structural analysis
3. Demonstration that overlapping epitopes may bind in different confirmations to the same HLA
4. Preliminary evidence that CD8 T-cell responses may be protective in a mouse model.

I therefore recommend the paper be accepted for publication in its current form. I want commend the authors on beautifully created figures that are easy to follow. I typically have lots of quirks with figures in papers I review and these were just done very well.

We thank the Reviewer for their generous and positive comments and agree with the comments about the importance of the findings.

Reviewer #2 (Remarks to the Author):

In the manuscript by Hensen et al. the authors define the influenza epitopes recognized by an Indigenous population in Australia that enriched in the HLA-A 24:02 allele that worldwide has a 10% global distribution. Already characterized epitopes in the general population as well as

the newly described epitopes are tested for their ability to evoke functional Th1 responses from PBMC and the frequency in Indigenous and non-Indigenous populations described. The manuscript is well written and thorough in its experimental approach. The authors argue that differences in immunodominant epitopes contribute to the increased susceptibility of the Indigenous population to severe influenza disease.

We thank the Reviewer for their positive feedback.

1. The manuscript may be more digestible if divided into two separate manuscripts. One defining the immunodominant epitopes in the two populations and their characteristics and another follow up discussing the functional implications.

We thank the Reviewer for this comment, however we believe, in line with Reviewer 1 and 4, that the manuscript as a whole is more informative as it combines descriptive results with functional implications.

2. Indigenous individuals display increased susceptibility to numerous other pathogens and APC have been implicated. Inclusion of a discussion of such information would add to the manuscript. As would inclusion of literature regarding the impact of previous infections on CD8 responses and disease outcomes.

We have made the following amendments to the manuscript to include additional susceptibility to other pathogens and the impact of previous infections of CD8⁺ T cell responses:

Page 10 (Results): “Changes in immunodominance after heterologous influenza virus infection have been studied previously, especially in B6 mice. Immunodominance hierarchy and cross-protective capacity of epitope-specific CD8⁺ T cells depend greatly on the first influenza encounter and presentation of viral epitopes by different cell types (reviewed in⁴⁴).”

Page 22 (Discussion): “*Non-epitope-specific binding of peptide/HLA-A24 complexes to KIR. This can possibly limit TCR recognition and thus TCR--specific activation of influenza-specific CD8⁺ T-cells, thus could provide a potential explanation to increased susceptibility to other infectious diseases such as sepsis and tuberculosis observed in Indigenous populations^{14,15}.*”

3. The LIFT cohort information in Figure 1b focuses on females. Both male and female information should be described, particularly for HLA-A24+ proportions in both sexes.

To clarify for the Reviewer, although Figure 1b describes the proportion of females included in our study, the frequency of A*24:02⁺ individuals was calculated for the total population, which is in line with the allele frequencies reported for the global and Indigenous populations from Figure 1a. No males were excluded in any data sets. Frequency of female donors was indicated for simplicity and to describe the cohort. Experiments were performed independently of the gender.

4. All representative Facs plots with gates should include frequencies and the shading on plots in figures (Fig 6 and 7) should be removed or lightened as the contours are difficult to see.

We thank the Reviewer for the suggestions. The shading on the FACS plots have been lightened for Figures 6d, 6f and 7c, and we have added the frequencies to the representative FACS plots for Figures 6a, 6d, 6f, 7c, 7d and 7g. An example of Figures 6a and 6d are shown below.

Modified Figure 6a

Modified Figure 6d

5. Clarity is required regarding the ungated seemingly tetramer positive cells in Figure 6a NP tetramer in the spleen and tonsil. Do these cells respond to peptide stimulation and become dual positive tetramer and cytokine producing cells? If these are real cells, the data would conflict with the main hypothesis of the manuscript.

Spleen and tonsil cells were not subjected to peptide stimulation, however, in our experience, CD8⁺ T cells from these tissues often bind HLA-A24 tetramers (not just the NP₁₆₅₋₁₇₃ tetramer) in a non-specific manner at a lower affinity compared to the true higher affinity antigen-specific population (Hensen, Kedzierska and Nguyen, unpublished findings). *Ex vivo* peptide stimulation and ICS were performed with PBMCs from Non-LIFT 8 donor showing no response (IFN γ and TNF) despite high frequency of tetramer-positive CD8⁺ T cells, indicating that these high frequency, low affinity tetramer-binding *ex vivo* populations do not respond to peptide. Additionally, we could show in Fig 7g that KIR3DL1 blocking prior to enrichment eliminated this low affinity tetramer-binding population from the unenriched sample.

Following the Reviewer's comment, we added the following sentence to the manuscript (page 18):

"*Ex vivo* peptide stimulation and ICS performed with PBMCs from Non-LIFT 8 donor showed minimal IFN γ /TNF response (data not shown), despite high frequency of tetramer-positive CD8⁺ T cells, indicating that these high frequency, low affinity tetramer-binding *ex vivo* populations were not peptide-specific".

6. The authors use numerous strains of influenza in mice and should provide information regarding their mouse adapted nature, especially for those that are not as common and well-described such as X31 and PR8.

IBV virus strains used in this study were human isolates grown in embryonated chicken eggs and have been described previously (Koutsakos et al. 2019, Nature Immunology). We have now included "human, non-mouse adapted isolates" in reference to the IBV strains on Page 35 in the Methods section.

7. In the protection experiments in Figure 5, the differences in weight loss are marginal and error bars overlapping. The authors could consider using whole protein versus CD8 peptide priming as the lack of helper epitopes will lead to the generation of helpless CD8 T cells and the findings of their formation and function as memory questionable.

We thank the Reviewer and agree that our vaccine model may not be ideal for achieving an optimal long-term protective effect for generating functional CD8⁺ T cell memory due to a lack of CD4⁺ T cell helper epitopes. We performed the peptide-immunisation study as a proof-of-principle experiment to show the protective capacity of HLA-A24-restricted CD8⁺ T cells. We found that weight loss was significantly reduced on 3 consecutive days (d4, 5 and 6) and we have shown mean and SD, rather than SE bars, which we have now clarified in the Figure legend (page 30). In addition, we observed increased viral clearance, which was significant on d7, as well as significantly reduced cytokine profiles in the vaccinated mice, which is of high importance given that hypercytokinemia is a key factor of severe influenza disease (Wang et al. 2013, PNAS; and reviewed by Liu et al. 2016, Cell Mol Immunol). Taken together, our data demonstrate an appreciable level of protection from novel HLA-A24-restricted IBV-derived CD8⁺ T cell epitopes using a well-established approach developed within our group (Koutsakos et al. 2019, Nature Immunology). We are currently working on improving our vaccination schemes and their protective effect but believe that whole protein immunisation raises other challenges such as differential epitope processing and reduced antigen availability, which would therefore need to be further studied. As peptide vaccination has been successfully used in our laboratory previously and was well established, we utilised this approach here and believe that our study delivers a proof-of-concept for a CD8⁺ T-cell-based vaccine.

8. Given that most humans are not mono chain for HLA alleles, whether or not one can make assumptions regarding dominant immunogenic epitopes using mono-chain HLA-transgenic mice is not clear. While these mice are valuable to allow one to say a response can be generated, it is questionable whether inferences about immunodominance and vaccine formulations in the absence of other competing alleles can be made.

We agree with the Reviewer that HLA-A24 transgenic mice are very valuable in identifying immunogenic epitopes following a live virus infection in a whole animal system. We acknowledge that there are limitations to using mono-chain HLA-transgenic mice and as such,

we can only infer on the immunogenic epitopes found in the context of HLA-A24, which were also screened and validated in humans (including all peptides) and take into account all HLA molecules from a number of different HLA-A24⁺ donors.

We therefore discussed the limitations of using transgenic mono-chain HLA-transgenic mice in Discussion (page 22):

“HHD-A24 transgenic mice are a valuable tool for identifying immunogenic epitopes following virus infection in an animal system *in vivo*. However, as there are limitations to using mono-chain HLA-transgenic mice, our study infers to the immunogenic epitopes in the context of HLA-A24 allele in mice. These CD8⁺ T cells were, however, further screened and validated in human PBMCs across several peptides presented by all HLA molecules from a number of different HLA-A24⁺ donors. Thus, our comprehensive analysis of peptide presentation ~~across mouse and human HLA-A24 models~~ and immunogenicity defines the candidate IBV and IAV peptides for a CD8⁺ T-cell-targeting vaccine in HLA-A24⁺ individuals.”

Reviewer #3 (Remarks to the Author):

This is an well performed study, that identifies previously unknown epitopes of influenza B that are the immunodominant in indigenous populations expressing the HLA A*24:02 allomorph. This result is not unexpected, since sufficient strength of binding of peptides to HLA alleles is essential for forming a T cell epitope. It is however significant because it reveals that focusing on just a few viral proteins may be a poor strategy for successfully immunizing diverse racial groups, when they express distinct HLA alleles. The studies re-enforces the fact that in most individuals only a few peptides are immunodominant, also well-known from previous studies, but since this may lead to lack of vaccine-induced protection in this case, it could have more stark consequences. Thus, this is an interesting study which indicates the impact of HLA variants in indigenous people on response to different influenza epitopes.

We thank the Reviewer immensely for their positive and valuable comments.

Detailed comments:

1. This is a thorough study with nice graphics. But it is very dense and it would be useful if the authors could give more summary information as they proceed.

We thank the Reviewer for the comments and appreciate the data rich nature of the manuscript. Where possible, and in keeping with the word limitations, we have endeavoured to provide a summary sentence at the end of each Results' sub-section. Table 1 also provides a list of the human immunodominant HLA-A24 epitopes, which we have now cited within the main text (page 12). We have also added an extra column to Supplementary Tables 2 and 3 to summarise which peptides are also immunogenic in other HLA or MHC molecules.

2. Is there more diversity in responses of non-indigenous populations or is the degree of restriction to immunodominant epitopes equivalent?

To address the Reviewer's comments, we refer to Figures 4a and 4b, which describes the diversity between responses in Indigenous and non-Indigenous populations. The diversity in responses were generally comparable across both groups having a similar number of immunodominant epitopes, although there were slight differences in terms of which epitopes

were immunodominant between the groups. For example, for IAV (Figure 4a), Indigenous responses (in red) were mainly directed towards PA₆₄₉₋₆₅₈ and PB1_{496/498-505} peptides, whereas non-Indigenous responses (in grey) were directed towards PB2₅₄₉₋₅₅₇ and PB1_{496/498-505} peptides, which is already described in the text (page 12 and shown below). These patterns of diversity were also observed towards IBV epitopes (Figure 4b), which we have now included a sentence as follows (Page 12): “*Diversity of CD8⁺ T cell epitopes was comparable between Indigenous (LIFT) and non-Indigenous (non-LIFT) PBMCs.*”

Current page 12:

“Interestingly, only 2/4 immunodominant epitopes observed in the Indigenous donors elicited comparably robust responses in 5 non-Indigenous donors screened (published epitopes: PB1₄₉₈₋₅₀₅ median 0.93% vs 1.1%, PB1₄₉₆₋₅₀₅ median 0.64% vs 0.8%), while the other two epitopes, PA₆₄₉₋₆₅₈ and NP₃₉₋₄₇ were poorly immunogenic in non-Indigenous donors who instead responded well to the PB2₅₄₉₋₅₅₇ epitope, absent in 4/5 Indigenous donors. Such differential epitope preference and immunodominance hierarchies between Indigenous and non-Indigenous donors is perhaps influenced by different HLA co-expressions or infection history.”

3. What evidence is there that vaccines currently in use fail to induce protection?

Following the Reviewer’s comment, we added the following sentences to Discussion (page 23):

“Current seasonal influenza vaccines are effective at inducing antibody responses to the currently circulating strains, however they do not protect against newly emerging pandemic viruses^{68,69}. We and others have previously shown that the inactivated influenza vaccines do not induce CD8⁺ T-cell immunity⁷⁰. Given the potential that CD8⁺ T cells have to protect against pandemic influenza viruses^{18,20}, vaccinations are clearly not harnessing the full power of the immune system, especially against pandemic viruses to which Indigenous populations are highly susceptible.”

4. There are a number of minor spelling errors.

We have corrected spelling errors.

Reviewer #4 (Remarks to the Author):

Hensen et al. characterize the HLA-24 associated influenza A and B virus immunopeptidome and associated CD8⁺ T cell response in humans and HLA-24 transgenic mice to better understand the high susceptibility of indigenous expressing individuals associated with HLA-24 expression. This is an outstanding “soups to nuts” study that exploits world class expertise of the collaborating investigators in mass spectrometry, structural biology, and human mouse immunology. The addition of so many peptides to the influenza peptidome, is in itself, a major contribution to the viral immunology field.

We thank the Reviewer for their generous and positive comments.

A few comments for the authors to consider.

1. Given the presence of at least 2, and up to 3 other HLA allomorphs in HLA-24 individuals, it seems more likely that any HLA-24 role in disease susceptibility is due not to a lack of function of HLA-24 in presenting peptides (as originally proposed by Hertz et al., 2013) but rather some deleterious gain of function effect. The identification of some many peptide ligands and T cells supports this interpretation. Perhaps the authors could raise this issue and briefly propose some alternative hypotheses.

We agree with the Reviewer that there is no deficit in the ability of HLA-A24 to elicit broad cross-reactive CD8⁺ T cell responses to IAV. We address other possible reasons for the role of HLA-A24 in disease susceptibility in Discussion, with amendments (underlined) added to further clarify these points:

Pages 20-21 (Discussion):

Hertz et al. previously showed that HLA-A24 has a low targeting efficiency for conserved regions of the pH1N1 virus, which was indicative of low cross-reactive memory responses that may have contributed to the impaired pH1N1 CD8⁺ T-cell immunity observed in HLA-A24⁺ individuals during the 2009 pandemic³⁰. Our data reveal that HLA-A24 presents a breadth of peptide ligands (52 peptides) derived from 6/8 IAV proteins. Notably, the variable HA and NA viral glycoproteins play a minimal role in HLA-A24-restricted CD8⁺ T-cell immunity to IAV. Instead, the focus on epitopes from PB1 and PB2 that are well conserved across virus strains circulating in South-East Asia and Australia suggests that the prominent HLA-A24 restricted CD8⁺ T-cell responses are likely to confer broad cross-reactive immunity to IAV. This is of key importance as the current T-cell vaccines in clinical trials focus mainly on structural proteins like NP, M1 and M2, and would therefore not elicit cross-protective CD8⁺ T-cell responses in HLA-A24⁺ individuals at risk of severe influenza disease.

Page 22 (Discussion): *'Our present study not only provides comprehensive data on generating CD8⁺ T-cell immunity against severe influenza disease in HLA-A24-expressing Indigenous and non-Indigenous people worldwide but also unravels three potential reasons why HLA-A24 has a role in disease susceptibility: (i) The antigenic origin of HLA-A24 IAV epitopes. The majority (62%) of IAV peptides presented by HLA-A24 are derived from PB1 and PB2, which is in stark contrast to previous studies in humans^{53,63,64,65} and mice⁶⁶ showing that immunodominant peptides for other HLAs are derived predominantly from NP, PA or M1. This is problematic for the current vaccine candidates in clinical trials which do not have a PB1 or PB2 component^{49,67,68}, and also raises the question of whether PB1 or PB2-specific CD8⁺ T-cell responses are equivalently robust and protective compared to immunodominant responses derived from NP, PA or M1 presented by other HLAs. (ii) Qualitative deficiencies in the HLA-A24 IAV-specific CD8⁺ T-cell response. In HHD-A24 mice, the magnitude and breadth of IAV-specific CD8⁺ T-cell responses were greatly reduced during secondary IAV (but not IBV) challenge compared to primary infection, implicating possible functional defects at memory establishment or recall levels. (iii) Non-epitope-specific binding of peptide/HLA-A24 complexes to KIR. This can possibly limit TCR recognition and thus TCR-specific activation of influenza-specific CD8⁺ T-cells. The above observations provide a platform for further investigations to understand the mechanisms driving greater risk of severe influenza disease in HLA-A24⁺ individuals.'*

2. It would be useful for non-specialist readers to provide a brief summary of CD8+ T cell immunodominance and a few references to relevant reviews.

We have amended the manuscript to include information of the key impact factors on CD8⁺ T cell immunodominance at first mention of a response hierarchy and provide a review on page 9 as follows:

“Several key determinants can orchestrate the main peptide target for CD8⁺ T cells and therefore the immunodominance of CD8⁺ T cell responses. These include the affinity of the peptide binding to the HLA I molecule, the affinity of the CD8⁺ T cells for an epitope, peptide availability (antigen dose) as well as the number of naïve epitope-specific CD8⁺ T cells (as reviewed in⁴³).”

3. No comments on the kinetics of peptide generation or the identification of out of reading frame peptides? Is there a relationship between peptide abundance and time of detection? The early detection of most peptides from biochemically stable proteins (all of the flu proteins except PB1-F2) strongly points to DRiPs being the major source of peptides, particularly since the frame shifted peptides appear in the same time frame as most other peptides. A few sentences at least would seem to be called for unless the authors plan expand this analysis in another paper.

We agree with the Reviewer that this is a very interesting finding raising the aforementioned questions. We are currently investigating this further, with a view to publish a follow-up manuscript.

4. At least one of the peptides identified, NP39 is known to be ligand for a different MHC I molecule (Kd). This ought to be mentioned and referenced, as well as any other peptides that are known to be presented by other allomorphs, as can be revealed simply by searching peptidome DBs.

We thank the Reviewer for their valuable comment. We have added a column to Supplementary Tables 2 and 3 to include whether our screened peptides were reported previously to be immunogenic and presented by other HLA/MHCs.

5. It should be mentioned that C1R cells are B cells that constitutive express immunoproteasomes and as such, likely have a highly overlapping but distinct peptidome compared to cells expressing standard/mixed proteasomes.

The Reviewer makes an important point and we have added the following underlined sentence to the discussion on Page 20:

‘Our in-depth mass-spectrometry-based immunopeptidomics approach defined the breadth of peptides presented by HLA-A*24:02 during IAV and IBV infection (Tables 1 and 2) and provided important insights into the characteristics of the associated CD8⁺ T-cell responses that could predispose to more severe influenza disease. It should be noted that the peptidomes identified by our approach are characteristic of C1R cells (derived from EBV-transformed B cells³³), which constitutively express immunoproteasomes. Although the presence of reactive T cells confirms the relevance of identified peptides, there is a possibility that some immunogenic peptides may be missing from our analysis due to tissue-specific processing.’

REVIEWERS' COMMENTS

Reviewer #2 (Remarks to the Author):

The authors have addressed my concerns.

Reviewer #3 (Remarks to the Author):

Overall impression: Their biochemical and structural studies seem to have been done pretty thoroughly. The functional studies, particularly in Figure 6, were all done at the surface level and have lots of holes in them. If all they want to say is that HLA-24 allele does have functional implications, these experiments suffice. But beyond that, they don't say much about phenotype, function or even best epitopes to design vaccines with. Below are my specific concerns:

1- In Figure 4B, they show that indigenous people have diminished responses to the various IBV epitopes compared to the non-LIFT group. Why then, in Fig 6B, are the precursor frequencies of the NP165-173 and NA32-40 IBV tetramer positive cells not any lower in indigenous compared to non-indigenous people? These two specific responses were markedly lower in Fig 4B. I wonder if it is because of only 2 indigenous IBV infected samples specific for NP and NA were analyzed in Fig 6. Would it then be misleading to draw a conclusion from the little data?

2- In Figure 6B, is it accurate to call CD8 tetramer frequencies in infected and convalescent samples "precursor"?

3- In Figure 6, how many days after infection was classified as acute vs convalescent?

4- In Figure 6d, are all acute and convalescent patient data pooled? One cannot pool CD8 effector and memory data from acute and convalescent phases since these are two different stages of the response.

5- In Figure 6d, it is very interesting that the memory subsets, both CM and EM are increased in indigenous groups. The authors do not discuss this. Would this then imply that indigenous people may have better memory responses? This would go against the authors' point in the discussion where they discuss that a lowering in the breadth of secondary IAV responses points to a defect in memory.

6- In Figure 6, the 'n' is low, often between 2 and 4 in many groups. Is that sufficient to draw conclusions from?

7- In Figure 5, how many times was the vaccination experiment repeated. The Figure shows that immunizing with IBV peptides provides some level of protection against IBV. This is expected. It would be very interesting if the authors could build on the novel findings in the paper: If they immunized with peptides that don't elicit a HLA-24 response in Figure 3, how does the protection compare to immunization with peptides that do elicit a HLA-24 response?

Reviewer #4 (Remarks to the Author):

Excellent job of positively responding to the reviews.

Responses to Reviewer 3:

Overall impression: Their biochemical and structural studies seem to have been done pretty thoroughly. The functional studies, particularly in Figure 6, were all done at the surface level and have lots of holes in them. If all they want to say is that HLA-24 allele does have functional implications, these experiments suffice. But beyond that, they don't say much about phenotype, function or even best epitopes to design vaccines with. Below are my specific concerns:

We thank the Reviewer for their appreciation of the biochemical and structural studies. We address Reviewer's additional comments below.

1- In Figure 4B, they show that indigenous people have diminished responses to the various IBV epitopes compared to the non-LIFT group. Why then, in Fig 6B, are the precursor frequencies of the NP165-173 and NA32-40 IBV tetramer positive cells not any lower in indigenous compared to non-indigenous people? These two specific responses were markedly lower in Fig 4B. I wonder if it is because of only 2 indigenous IBV infected samples specific for NP and NA were analyzed in Fig 6. Would it then be misleading to draw a conclusion from the little data?

The data in Fig. 4B show expansion of CD8⁺ T cells after 15 days of *in vitro* culture. Fig 4B shows that the expansion capacity of the CD8⁺ T cells is highly variable between individuals. A variety of different reasons could be responsible for this, including co-expression of other HLAs, age of the individual or infection history. In Fig. 4B, we show that while the median of responses was lower in the Indigenous donors, the magnitude of response was still comparable between responses in Indigenous and non-Indigenous donors. As the samples used for TAME were different to the samples used for *in vitro* expansion (given the limited PBMC number from particular donors), we think it would not be appropriate to draw any direct conclusion without strong evidence. One reason for the reduced expansion in the Indigenous samples could be the expression of specific KIR receptor. However, this needs further analysis which is beyond the scope of this publication. We are working on a follow-up publication and have included this in the Discussion:

Discussion (p21)

“Non-epitope-specific binding of peptide/HLA-A24 complexes to KIR. This can possibly limit TCR recognition and thus TCR-specific activation of influenza-specific CD8⁺ T-cells and could also provide an explanation for increased susceptibility to other infectious diseases such as sepsis and tuberculosis observed in Indigenous populations^{14,15}.

2- In Figure 6B, is it accurate to call CD8 tetramer frequencies in infected and convalescent samples “precursor”?

Following the Reviewer's comment, we deleted the word 'precursor'.

3- In Figure 6, how many days after infection was classified as acute vs convalescent?

We thank the Reviewer for this comment. Donors acutely infected with influenza were within 10 days of disease onset, whereas convalescent donors were sampled 30 days post disease onset. We have modified the text to include this information.

4- In Figure 6d, are all acute and convalescent patient data pooled? One cannot pool CD8 effector and

memory data from acute and convalescent phases since these are two different stages of the response.

We do agree with the Reviewer that memory phenotypes transit between acute and convalescent donors. The main message from this figure is the reduction of a naïve phenotype and an increase of a memory (CD45RA⁺; T_{EM} or T_{CM}) phenotype, which holds up for both acute and convalescent donors. Grouping the data allows us to strengthen the statistics.

5- In Figure 6d, it is very interesting that the memory subsets, both CM and EM are increased in indigenous groups. The authors do not discuss this. Would this then imply that indigenous people may have better memory responses? This would go against the authors' point in the discussion where they discuss that a lowering in the breadth of secondary IAV responses points to a defect in memory.

The age of the Indigenous participants whose blood was used for the TAME was relatively low (21-27 years), while the age of the non-Indigenous donors was variable (24-60 years), which might be one possible reason for these observations (Supplementary Table 1). Therefore, based on our data, we cannot conclude that Indigenous people have better memory responses.

6- In Figure 6, the 'n' is low, often between 2 and 4 in many groups. Is that sufficient to draw conclusions from?

We do not draw any strong conclusions from data with low n-values. These studied tissues are difficult to source, and we believe that the inclusion of *ex vivo* data strengthens the studies and highlights the involvement of the identified epitope-specific CD8⁺ T cells in combating influenza virus infections.

7- In Figure 5, how many times was the vaccination experiment repeated. The Figure shows that immunizing with IBV peptides provides some level of protection against IBV. This is expected. It would be very interesting if the authors could build on the novel findings in the paper: If they immunized with peptides that don't elicit a HLA-24 response in Figure 3, how does the protection compare to immunization with peptides that do elicit a HLA-24 response?

The experiment was performed once as a proof of principle given the high number of mice required, paired with difficulties to breed HHD-A24 mice. However, this experiment has been performed previously in our laboratory with comparable outcomes in an HHD-A2 model (Koutsakos et al. 2019 PMID: 30778243, Nature Immunology). Furthermore, previous preliminary experiments in our laboratory using intranasally administered unspecific lipopeptides towards peptides presented by HLA-A2 (A2/M1₅₈) in HLA-A24 mice showed lack of protective immunity by A2/M1 peptide, suggesting that injection of unspecific peptides do not induce protection against influenza virus infection.

Reviewer #4 (Remarks to the Author):

Excellent job of positively responding to the reviews.

We thank the Reviewer for appreciating our efforts.